

# 1 Multiscale Modeling for Coastal Cities: Addressing Climate Change
# 2 Impacts on Flood Events at Urban-Scale

Michele Bendoni[1], Francesca Caparrini[2], Andrea Cucco[3], Stefano Taddei[4], Iulia Anton[5], Roberta Paranunzio[6], Rossella Mocali[4],
Massimo Perna[4], Michele Sacco[4], Giovanni Vitale[8,4], Manuela Corongiu[4], Alberto Ortolani[7,4], Salem Gharbia[5], Carlo Brandini[8,4].
1. Institute of Marine Science, National Research Council of Italy (CNR-ISMAR), Forte Santa Teresa, snc, 19032 - Lerici (SP),
Italy.
2. Institute of Geosciences and Earth Resources, National Research Council of Italy (CNR-IGG), Via G. Moruzzi 1, 56124  - Pisa
(PI), Italy.
3. Institute for the study of Anthropic Impacts and Sustainability in marine environment, National Research Council (CNR- IAS),
Loc. Sa Mardini Torregrande - Oristano, Italy.
4. LaMMA Consortium, Via Madonna del Piano 10, 50019 Sesto Fiorentino (FI), Italy.
5. Atlantic Technological University, Ash Lane F91 YW50, Sligo, Ireland.
6. Institute of Atmospheric Sciences and Climate, National Research Council of Italy (CNR-ISAC), Corso Fiume, 4, 10133 Torino
(TO), Italy.
7. Institute of Bio-Economy, National Research Council of Italy (CNR-IBE), Via Madonna del Piano 10, 50019 Sesto Fiorentino
(FI), Italy.
8. Institute of Marine Science, National Research Council of Italy (CNR-ISMAR), Via Madonna del Piano 10, 50019 Sesto
Fiorentino (FI), Italy.
Corresponding Author: Carlo Brandini, brandini@lamma.toscana.it, https://orcid.org/0000-0002-6509-4533
**Abstract**
This study presents an integrated modeling framework designed to bridge scales from regional to urban,
enabling a detailed assessment of the impacts of future climate scenarios on three European coastal cities:
Massa (Italy) and Vilanova (Spain) in the Mediterranean, and Oarsoaldea (Spain) in the Atlantic. Conducted
as part of the SCORE EU Project (*Smart Control of Climate Resilience in European Coastal Cities*), the
framework employs a novel, non-standard downscaling approach to translate large-scale atmospheric
outputs from the EURO-CORDEX regional model ALADIN63 (for Historical, RCP4.5, and RCP8.5
scenarios) into high-resolution simulations of storm surges, wave climate, and river discharge using
SHYFEM, WAVEWATCH III, and LISFLOOD models.
The framework achieves coastal resolutions on the order of 100 m, providing time series of water levels
and wave runup, which are combined into total water levels. These results, together with extreme value
analysis of river discharge and projected relative sea level rise (RSLR), are used as boundary conditions for
an urban-scale hydrodynamic model with resolutions as fine as 2–20 m. This multi-scale integration allows
for detailed analysis of changes in flooded areas and volumes under RCP4.5 and RCP8.5 scenarios, relative
to historical conditions, highlighting the influence of shifting extremes, RSLR, and site-specific features.
Results show that in Massa and Vilanova, increased extreme river discharges are projected, while moderate
changes in extreme water levels are overshadowed by RSLR, particularly for Massa. Oarsoaldea, well
protected from storm surges, is expected to experience a slight reduction in extreme river discharge. This
work demonstrates the capability of the integrated framework to address climate change impacts at urban
scales, providing valuable insights for the development of localized adaptation strategies.





## 1 Introduction

Rapid urban growth and climate change are two of the most pressing challenges of our time (Satterthwaite, 2009), especially in coastal regions, where their combination significantly increases the exposure of urban areas to extreme natural events. Coastal cities and settlements, home to more than 2 billion people worldwide, are among the most vulnerable areas to these events (IPCC, 2023; Vitousek et al., 2017; Oppenheimer et al., 2019). Approximately 900 million people live in low-elevation coastal zones (LECZ), areas situated less than 10 m above mean sea level (Reimann et al., 2023), with a projected global population density of around 400-500 people/square km by 2060 (Neumann et al., 2015). These regions, marked by increasing anthropogenic activity, hold crucial social and economic importance, with dense population and infrastructure that may further elevate their future vulnerability (Figueiredo et al., 2024; Paranunzio et al., 2022). Global mean sea level is projected to rise between 0.3 and 2 m by 2100 under scenarios of increasing global warming (Vitousek et al., 2017). In addition, the effects of land subsidence are expected to further exacerbate risks in most coastal areas, intensifying future impacts on population and infrastructure (Vousdoukas et al., 2018).

In Europe alone, currently, over 50 million live in LECZ areas (Vousdoukas et al., 2020). With a relative sea level rise (RSLR) of just 0.15 m above 2020 levels, the coastal population potentially exposed to a 100-year coastal flood could increase by about 20% in the medium to long term (IPCC, 2023). By 2100, the total number of people exposed to risk of flooding is projected to reach 1.61 million, and 3.9 million, under the two Representative Concentration Pathways (RCP) scenarios 4.5 and 8.5 (Vousdoukas et al., 2020).

Coastal cities around the world are threatened not only from inundation due to storm surges or sea level rise (Hallegatte et al., 2013; Wahl et al., 2017) but also from river flooding which poses additional risk (Khanal et al., 2019). These areas are therefore impacted by a complex interplay of multiple flood-related systems including river, sea/oceans and coastal land (Laino et al., 2024). Assessing the local effects of such hazards to enhance coastal communities' resilience is one of the greatest challenges of our time, especially in the context of the ongoing climate change. High uncertainty in urban sprawl and flood risks leads to a generalized lack of preparedness to face future flood events (Sun et al., 2022). In this context, high-resolution climate data are essential for defining downscaling strategies that begin with global climate services and are able to evaluate the impacts of multiple hazards at the local scale. Bensi et al. (2020) provides a broad overview of existing literature on hazard interaction, organized by different flooding hazard focus, i.e., studies that address several mechanisms in the fluvial and coastal flood processes alone and studies focusing on joint fluvial and coastal flood processes (e.g., Masina et al., 2015; Bevacqua et al., 2017). Many studies address the degree of dependence among different mechanisms, e.g., precipitation, river flow and storm surge events to assess coastal flood risk, also investigating how it changes over time (Bevacqua et al., 2017; Moftakhari et al. 2017; Orton et al., 2018; Zheng et al., 2013) and with respect to different climate change scenarios (e.g., Parodi et al. 2020; Zhong et al., 2023; Gori & Lin, 2022; Wahl et al., 2015).

Despite the large number of methodologies, tools and models exploring the single or combined effect of climate-related hazards in coastal areas worldwide, studies which exploit different approaches to provide a global multidisciplinary framework to assess flood scenarios in the future at the fine resolution of the urban scale are not widespread (Bensi et al., 2020). Some promising studies pointing in this direction have been developed during the last decade, especially in the US. Based on copulas and bivariate dependence analysis, Moftakhari et al. (2017) quantified the increases in failure probabilities of coastal flood defenses for eight estuarine systems along the coasts of United States caused by RSLR under multiple flood drivers ad RCP4.5





and RCP8.5 in 2030 and 2050. To assess climate impacts for the US West Coast, Barnard et al. (2014) used
wind fields from different Global Circulation Models (GCMs) under two RCPs scenarios, 4.5 and 8.5, to
resolve 3 hours peak conditions into the WAVEWATCH III wave models within a deterministic,
multidimensional framework in the Coastal Storm Modeling System (CoSMoS). Process-based modeling
system proved to be able to dynamically transfer information from global atmospheric scale to the regional
and local scale to predict impacts of multiple coastal hazards (i.e., coastal erosion and cliff failures and
flooding) for a range of RSLR and storm scenarios at a resolution scale that is relevant for management and
adaptation planning (meters scale) (Barnard et al., 2019). In Europe, some few attempts have been made to
develop comprehensive models that scale down from the synoptic to the urban scale. Model framework to
assess the coastal risks and morphological impacts induced by extreme storm events similar to CoSMoS
has been developed in the context of European projects (e.g., Ciavola et al., 2011), but more in support of
early warning and emergency response. Van den Hurk et al. (2015) studied the joint distribution of
precipitation and storm surges for 1950 to 2000 using 800 years of simulated data using a RACMO2
Regional Circulation Model (RCM) at 12 km resolution to establish a relation between compound hazards
in the Netherlands.
It follows that high resolution RCMs are needed to properly model climate impact at a higher resolution.
Estimating the impacts of climate change on coastal cities requires increasing the resolution of city-scale
models to unprecedented levels, simulating coastal and terrestrial flood conditions for different return
periods and scenarios, and including considerations for the evaluation of financial resilience strategies or
ecosystem-based adaptation solutions. Thus, a multidisciplinary framework is needed to foster, through co-
participatory and co-creative approach, the public engagement of scientists, policy-makers and citizens, to
identify and share socially and technically acceptable solutions. This is part of SCORE project (Smart
control of climate resilience in European coastal cities, https://score-eu-project.eu/) which aims, through an
integrated and multidisciplinary approach, to monitor and validate reliable and robust adaptation measures
in low-lying coastal cities to minimize the effects of climate-related hazards and enhance the overall
resilience. This is addressed in the context of the Coastal City Living Labs (CCLLs), a novel participatory
approach built upon the living lab concept that aims to involve scientists, decision makers, citizens and
different stakeholders in the modeling process and in preparing climate risk assessment analysis, thus
accelerating the systematic adoption (Paranunzio et al., 2023).
To assess the impacts of multiple climate-related hazards on coastal cities under different climate change
scenarios, we present a downscaling procedure which consists of a dynamic multi-branch modeling chain
ending with high-resolution (~2 m) flood simulations. Here, we use the term "downscaling" to indicate the
transfer of information from the synoptic atmospheric scale to the urban scale of individual buildings and
streets, rather than the increase in detail of a specific dataset coming from a numerical model with higher
spatial and temporal resolution with respect to the parent one. An integrated approach blending
oceanography, hydrology, hydraulics and extreme value analysis (EVA) has been used for the computation
of flooded areas for both historical periods and future climate projections for different return periods and
under two different RCP scenarios, 4.5 and 8.5 (IPCC, 2014). We used atmospheric data from an EURO-
CORDEX RCM (Jacob et al., 2014), and three different models simulating the evolution of water level,
wave dynamics, and rainfall-runoff transformation to create the boundary conditions to run hydrodynamic
simulations in coastal cities, for both past and future periods. The modeling chain has been applied to the
three different CCLLs based on the indications of the SCORE Project: Massa (Italy), Vilanova i la Geltrù
and Oarsoaldea (Spain), as different test cases characterized by different phenomenological features.





The high computational demand of the simulation and the need for an extremely fine temporal resolution
data are two major challenges in this context. Among the EURO-CORDEX models, only one RCM offers
at least three-hourly data for the atmospheric variables required across all models and scenarios. We
acknowledge that the use of a multi-RCM (GCM) ensemble is preferable with respect to a single RCM
(GCM) to predict more rigorously spatial patterns and to estimate the uncertainty in the projections in
response to climate change (Khanal et al., 2019; Gori & Lin, 2022; Bevacqua et al, 2020; Ghanbari et al,
2021). However, the computational cost of the procedure and the high-resolution of the model create
challenges for multi-model impact assessment at urban scale. In addition, some studies make successful use
of one GCM in dynamical downscaling and hydrological modeling (Vezzoli et al., 2015; Lima et al., 2023).
To our knowledge, this is one of the first works for the European area dealing with projections of climate
data at i) such a high spatio-temporal resolution, ii) exploiting various computational demanding models
up to the urban scale, iii) seeking to develop a flood hazard modeling chain from multiple sources and iv)
embracing a multidisciplinary modeling framework.
The work is organized as follows. Section 2 provides a brief overview of the project and description of the
study sites. Section 3 describes the overall methodology, while Section 4 deals specifically with the
implementation of the three numerical models. Section 5 describes the extreme value analysis and the urban
scale model. Results of the overall methodology are then presented in Section 6  and discussed in the next
section. Section 8 is dedicated to conclusion on outlook.

## 2 The SCORE Project and the study sites

The SCORE project focuses on the resilience of coastal cities to the effects of climate change. Coastal
cities, as climate change hotspots, are affected by numerous consequences resulting from changes in the
marine, atmospheric, and terrestrial (hydrogeological) components of the Earth system. However, among
the many risks related to climate change in coastal cities (which could include increasing marine and
atmospheric heatwaves, fire risks, subsidence due to the over-exploitation of water resources in tourist
areas, etc.), SCORE has focused on flood risk. This includes flooding from rivers, marine inundations, or a
combination of both. Marine floods, as is well known, can result not only from extreme storm surges but
from combinations of storm waves and high tidal levels (both astronomical and meteorological induced by
wind and pressure), following a signal that is modulated in the long term by RSLR.
The selection of cities involved in the project was made during the project development phase. The choice
was not driven by prioritizing cities with the highest exposure to these effects (e.g., the city of Venice), but
rather those where there is an active and engaged community of citizens, stakeholders, and research centers
collaborating on co-designing solutions to improve resilience to the effects of climate change. This process
begins with ecosystem-based adaptation solutions (EbAs; Munang et al., 2013; Temmerman et al., 2013;
Tiwari et al., 2022), which encourage practices that increase citizen participation and awareness, such as
sharing meteorological observations following Citizen Science standards (Conrad & Hilchey, 2010). The
modeling components developed for these cities also contribute to the creation of urban-scale Digital Twins,
which are part of a specific activity within the project. These digital tools, alongside advanced data
representation, enable a better understanding of flood effects and allow the modeling of adaptation scenarios
using a What-If methodology (Paranunzio et al., 2023).
Within the project, local initiatives are built following the Living Lab paradigm (Bulkeley et al., 2018),
forming Coastal Cities Living Labs, where local communities participate according to the quadruple helix



model (Carayannis & Campbell, 2009). The decision of whether cities would act as frontrunners or
followers for certain project activities (as organized through the project work packages) was made based
on the specific themes of interest within the CCLLs.
Therefore, the selection of the study cases presented in this article: Massa, Vilanova i la Geltrù (from now
on we will refer to the city simply as Vilanova), Oarsoaldea (Figure 1) was based on the presence of three
frontrunners that followed a common analysis methodology, which is described in the next section. This
methodology starts from the availability of data provided by climate services and, through downscaling
techniques and urban and coastal hydraulic modeling, defines the design conditions expected for coastal
cities. Defining case studies based on project guidance does not diminish the scientific value of this work
or the approach used; rather, it demonstrates how the problem of coastal resilience is universal and not
restricted to specific areas. Ultimately, this requires a careful analysis that can be more effectively carried
out with a local and site-specific approach rather than relying solely on regional models, even when they
have high-resolution.

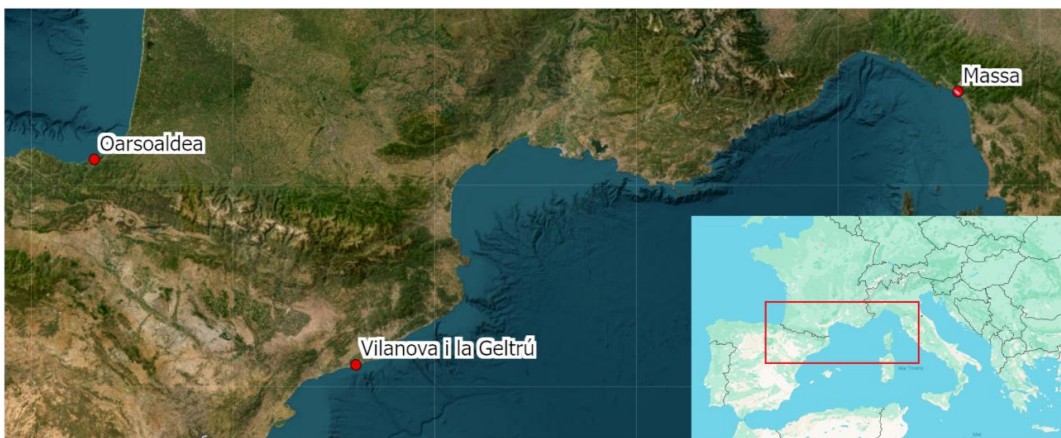

**Fig 1** View of the geographical area where the analyzed cities are located. Base map: Google Satellite imagery (©
Google 2024; Imagery © CNES / Airbus, Maxar Technologies, Airbus)

**3 Overall methodology**
The modeling chain implemented transfers information from the atmospheric synoptic scale (1000-100 km)
up to the urban scale (2 m), and is aimed at obtaining time series of wave height $H_s$, water level $\eta$, and river
discharge $Q$ close to the coastal cities of interest, for both past periods and future climate projections. An
extreme value analysis is then performed on the calculated time series to estimate the peak values associated
with specific return periods. These values are eventually employed to build synthetic events to simulate
their effects in terms of flooded areas for the analyzed coastal cities. A sketch of the overall procedure is
reported in Figure 2.



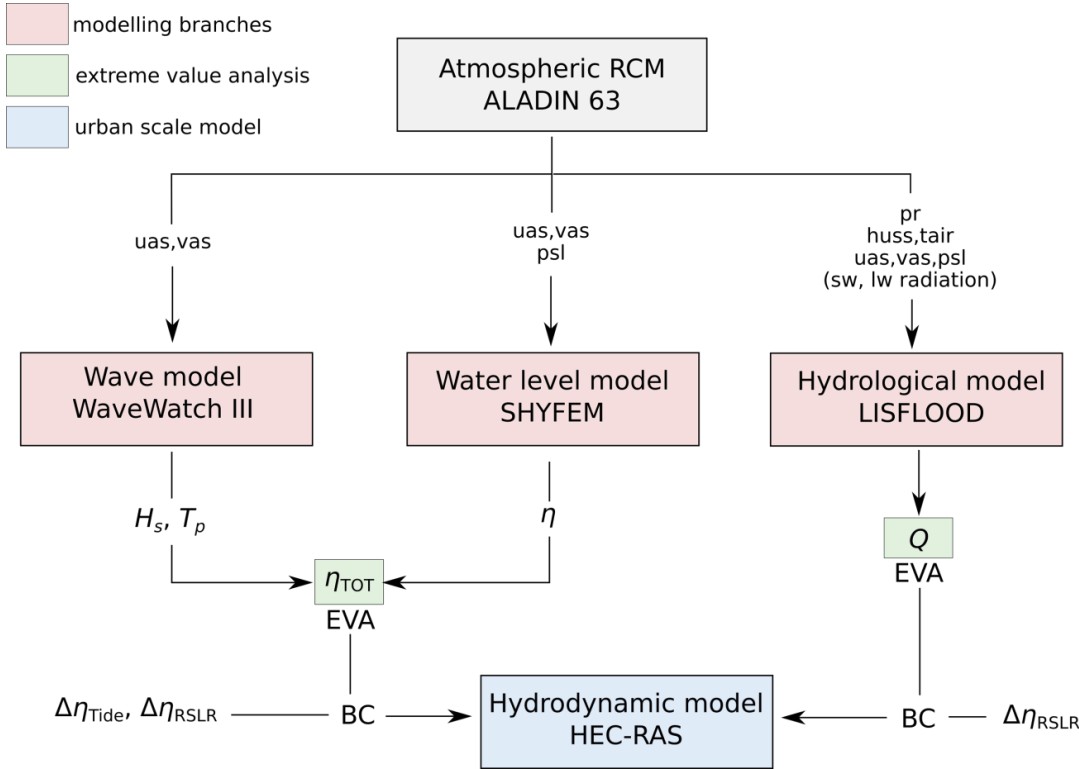

**Fig 2** Sketch reporting the overall methodology to downscale data and run hydrodynamic simulations at the urban scale. Light red boxes correspond to the models employed to downscale atmospheric variables, light green boxes contain variables subject to extreme value analysis and the light blue box corresponds to the urban scale flood modeling part. $H_s$ is the significant wave height, $T_p$ is the peak wave period, $Q$ is the river discharge, $\eta$ is the water level, $\Delta\eta_{Tide}$ and $\Delta\eta_{RSLR}$ are the increases in water level due to tide and relative sea level rise, respectively

The modeling chain implemented employs atmospheric data from the ALADIN63 RCM (Coppola et al., 2020; Vautard et al., 2020), provided by the EURO-CORDEX experiment (Jacob et al., 2014), and use it as input for the following models: WaveWatch III (WW3DG, 2019) simulates the dynamic of wave height taking as input the surface zonal and meridional wind velocities (uas, vas); SHYFEM (Umgiesser et al., 2004) simulates the evolution of water levels forced by surface winds (uas, vas) and mean sea level pressure (psl); LISFLOOD (Van Der Knijff et al., 2008) simulates the rainfall-runoff transformation and takes in input several atmospheric variables such as rainfall rate (pr), air temperature (tair), specific humidity (huss), sea level pressure (psl) shortwave and longwave radiation (rsds, rlds, rsus, rlus). A more detailed and thorough description of the downscaling procedure for each variable is reported in Sections 4.1, 4.2 and 4.3.

For each of the models, the Evaluation, Historical, RCP4.5 and RCP8.5 experiments are simulated. The Evaluation (Eval) experiment is employed to test the ability of the model to reproduce observable extreme events. In such a case the ALADIN63 RCM is forced by the ERA-Interim reanalysis (Dee et al., 2011). The Historical (Hist) experiment is used as a baseline for the two climate change scenarios expressed by the Representative Concentration Pathways defined by the fifth Assessment Report (AR5) of



Intergovernmental Panel on Climate Change (IPCC, 2014). RCP4.5 and RCP8.5 data are used to analyze
the effect of anthropogenic climate change in the future flooding pattern at urban scale. For this set of
simulations, the ALADIN63 RCM was forced by the CNRM-CM5 GCM (Voldoire et al., 2011). The choice
of such a RCM is due to the fact that this was the only one that provided at least three-hourly data for the
atmospheric forcing variables for all the experiments, among the EURO-CORDEX models. Other RCMs
provided those variables at different output frequencies or solely for specific temporal windows (e.g.
RCP4.5 for the period 2050-2070 and RCP8.5 for the period 2030-2050). The consequences and limitations
of such a choice are discussed in Section 7.
A summary of the simulated experiments with associated time windows is reported in Table 1.

| Experiment | Time window | Simulated RP [years] |
|:---:|:---:|:---:|
| Eval | 1980-2012 | - |
| Hist | 1956-2005 | 25, 100 |
| RCP4.5 | 2006-2100 | 25, 100 (2011-2060)<br>25, 100 (2051-2100) |
| RCP8.5 | 2006-2100 | 25, 100 (2011-2060)<br>25, 100 (2051-2100) |

Table 1. Summary of the simulated experiments with associated time windows. Return periods (RP) refer to the values calculated
through the extreme value analysis and used to create synthetic events simulated with the urban scale hydrodynamic model.

The hydrodynamic simulations of storm surges and river flood at urban scale have been performed using
the HEC-RAS 6.4 model (Brunner & US Army Corps of Engineers, 2021), similarly to Gori and Lin (2022).
The storm surge is modeled following a simplified approach consisting of the combination of time series
of wave runup $R_{2\%}$ and water level. First, the wave runup $R_{2\%}$ is determined using wave height and period
and the slope of the beach, following Atkinson et al. (2017), then, it is added to the water level $\eta$, to obtain
the total water level $\eta_{TOT}$. The extreme value analysis is carried out on this last variable and on the river
discharge Q, separately, for all the simulated experiments (Table 1). Hazard maps reporting the water depth
envelope associated with a specific return period event are produced for the flood due to the storm surge
and for the riverine flood. Furthermore, to simulate the RSLR and the effect of the tide, an increased value
for the mean water level is applied to each hydrodynamic simulation based on the associated experiment.
A more detailed description of the urban scale hydrodynamic modeling activity is reported in Section 5.
The projections of RSLR for RCP4.5 and RCP8.5 used in this paper can be found in two free-access datasets
(Vousdoukas et al. 2016a for RCP4.5 data, Vousdoukas et al. 2016b for RCP8.5 data), downloadable from
the European Commission Joint Research Centre (JRC) website. These datasets provide the Total Water
Level (TWL), from which the RSLR can be extracted by subtracting the episodic extremes (wave runup
and storm surge level) which are also provided, along with the tidal contribution. More information can be
found in the related article (Vousdoukas et al. 2017). The dataset covers the European coastlines with a
temporal resolution of 10 years. Europe is divided into 10 regions, within which all values are averaged.
All values are given with respect to the 1985–2005 reference period.



## 4 Modeling branches

In this section, we describe the implementation of the three numerical models: WaveWatch III, SHYFEM and LISFLOOD, employed to perform the main part of the downscaling procedure. Each of the models has a particular setup on the basis of the analyzed coastal city.

### 4.1 Wave climate model

The numerical model used to simulate wind waves was WaveWatch III (WAVE-height, WATer depth and Current Hindcasting), v. 6.07 (WW3DG, 2019), a community third-generation wave model developed at the US National Centers for Environmental Prediction (NOAA/NCEP) that includes the latest scientific advancements in the field of wind-wave modeling and dynamics (https://github.com/NOAA-EMC/WW3/releases/download/6.07/wwatch3.v6.07.tar.gz).

WAVEWATCH III solves the random phase spectral action density balance equation for wavenumber-direction spectra, and includes options for shallow-water applications. Propagation of a wave spectrum can be solved using regular (rectilinear or curvilinear) and unstructured (triangular) grids. Source terms for physical processes include parameterizations for wave growth due to the actions of wind, nonlinear resonant wave-wave interactions, scattering due to wave-bottom interactions, triad interactions, dissipation due to whitecapping, bottom friction, surf-breaking, and interactions with mud and ice. Source terms are integrated in time using a dynamically adjusted time stepping algorithm.

In this application, according to the project needs, two different implementations of the model were performed, with two different computational domains. The first one included the entire Mediterranean basin and a further area west of the Strait of Gibraltar, to improve accuracy in the Alboran Sea (Figure 3b). The second one was extended to the Atlantic Ocean (Figure 3a) to simulate the wave climate in front of the ocean-facing European cities. As for boundary conditions, domains were assumed to be closed at the farthest ocean boundaries. Both domains have been discretized by unstructured meshes with a variable resolution up to 500 m in the coastal areas surrounding the cities of interest (Figures 3c, and 3d). The resolution decreases in the rest of the domain and the minimum resolution in deep offshore areas reaches about 70 km for the Mediterranean grid, and about 300 km for the Atlantic one. GEBCO, EMODnet, and nautical chart bathymetries were used in different parts of the domains.

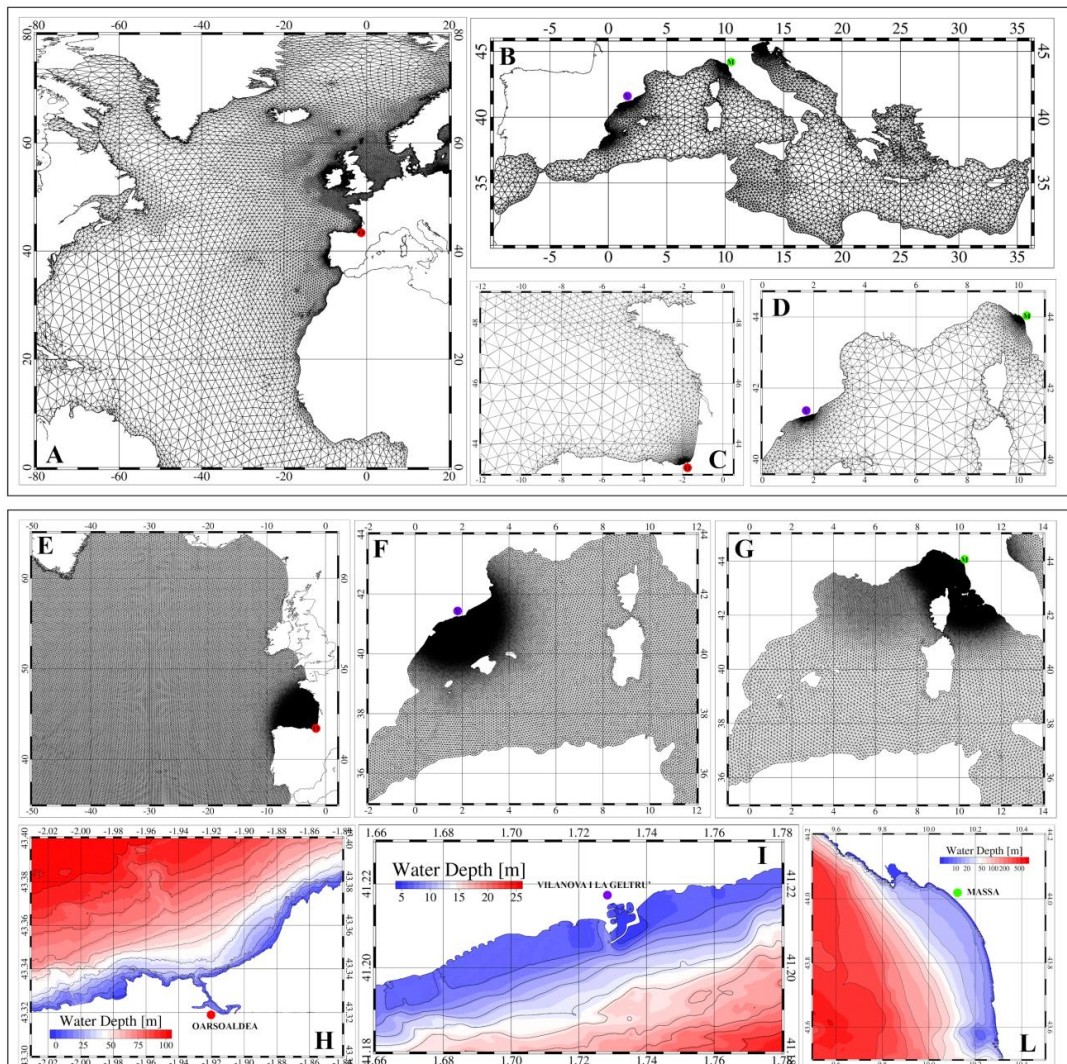

**Fig. 3** Finite element meshes used by the WWIII wave model (upper panels labeled with A, B, C and D) and the SHYFEM hydrodynamic model (lower panels labeled with E, F, G, H, I and L). Panels A and B show portions of the WWIII domains, which include most of the Atlantic Ocean and the entire Mediterranean Sea. High-resolution areas for Oarsoaldea (red point), Vilanova (blue point), and Massa (green point) are displayed in panels C and D. Panels E, F, and G illustrate portions of the three SHYFEM domains, covering most of the North Atlantic Ocean and the entire Mediterranean Sea, highlighting the high-resolution areas. The bottom panels (H, I and L) depict the bathymetric details of the three study sites

The output of the wave model was recorded hourly at all grid points for the integrated quantities, in particular significant wave height ($H_s$), mean wavelength ($L_m$), mean wave period ($T_m$), peak wave period ($T_p$), mean wave direction ($Dir_m$) and peak wave direction ($Dir_p$). The atmospheric dataset provided by ERA



Interim+EuroCordex (ALADIN63 RCM) for the evaluation data and CMIP5+EuroCordex for the other
data, which includes wind (uas, vas) at a frequency of 3 hours, was used as forcing.
**4.2 Water level model**
Future projections of storm surge events for the three study sites have been conducted using advanced
numerical modeling techniques. Specifically, SHYFEM (System of Hydrodynamic Finite Element
Modules, Umgiesser et al., 2004), an ocean model based on the finite element method, has been
implemented for each coastal site to simulate the temporal and spatial variability of water levels influenced
by atmospheric forcing, wind and atmospheric pressure.
SHYFEM is an open-source community model (freely downloadable at https://github.com/SHYFEM-
model/shyfem.git), that resolves the 3D primitive equations system, integrated over z-layers, in their
formulations with water levels and transports. It uses a semi-implicit algorithm for the discretization in time
and finite element for the spatial integration. The model has been widely used to investigate the main
hydrodynamics in coastal areas (e.g. Western Mediterranean Sea in Bonamano et al., 2024 and Cucco et
al., 2023, 2022; Umgiesser et al., 2014, 2022; Quattrocchi et al., 2021; Maicu et al., 2018; Federico et al.,
2017) and for real time prediction of storm surge events in several coastal sites in the Mediterranean sea,
e.g. the Venice Lagoon ( Umgiesser et al., 2022; Bajo et al., 2007, 2019). We refer to (Umgiesser et al.,
2004) for a detailed overview of the model equation system, numerical treatment and parameters setup.
In this application, SHYFEM has been implemented in 2D mode accounting for barotropic pressure
gradients, wind drag and bottom friction, which are the primary forces driving the storm surge events
(Bloemendaal et al., 2018; Wicks et al., 2017). The model was applied to simulate the atmospheric
contribution to water level $\eta$, thus neglecting the non-linear interaction with tides. This approach is
commonly used in ocean prediction systems, in fact, the non-linear interactions between tides and surge are
generally small enough to allow for the linear addition of tidal and surge components thus reducing the
complexity of numerical experiments (Yang et al., 2023; Zijl et al., 2013; Bajo et al., 2007).
The water levels including tides can be derived by adding the astronomical tide to the computed $\eta$. The
impact on accuracy depends on tidal amplitudes, which are minimal in the Western Mediterranean Sea due
to very low tides (0.2-0.3 m) and slightly more significant for the Atlantic site where tidal amplitudes exceed
1.5 m (around 3 m, as estimated by Fernández-Montblanc et al., 2018 for the whole European coastal seas).
The same assumption was applied to other factors such as general circulation and climate-induced RSLR,
which contribute to a lesser extent to water level fluctuations in case of extreme events.
Three different finite element meshes have been implemented to reproduce, with varying spatial resolution,
the geomorphological features of the three coastal sites (Figure 3h, i, l). Each domain extends to the entire
basin facing each study site (the Western Mediterranean Sea for Villanova and Massa, and most of the
North Atlantic for Oarsoaldea) to cover the full area influenced by the main wind fetches and to eliminate
the need for ad hoc open boundary conditions.
The atmospheric dataset provided by the ALADIN63 RCM, which includes wind and atmospheric pressure
data (uas, vas and psl) at a 3-hour frequency, was used as forcing.
**4.3 River discharge model**
River floods occur when the stream or channel geometry is not sufficient to contain the incoming volume
of water. In order to model river floods, it is necessary to define the discharge hydrographs, i.e. the evolution
in time of flow rate in given cross sections. The shape of the hydrograph, the time and value of its peak,
and in general the streamflow generated in the channel network as a response to precipitation events, are





the consequences of the hydrological processes in the upstream basin. Such processes include several
complex mechanisms occurring at land surface (infiltration, evapotranspiration, runoff generation, hillslope
routing, snowmelt, groundwater recharge) that depend on many factors like basin topography, soil hydraulic
properties, vegetation cover and structure of the river network.
In this work, we have used LISFLOOD (https://ec-jrc.github.io/lisflood/), a spatially distributed
hydrological model developed by the Joint Research Centre (JRC) of the European Commission since 1997
(Van Der Knijff et al., 2008). LISFLOOD has been applied to a wide range of applications and is currently
used in the EFAS (European Flood Awareness System) and GLOFAS (Global Flood Awareness System)
(Alfieri et al., 2019). In LISFLOOD, the soil is schematised with three layers and all the main hydrological
processes are modeled: surface runoff, exchange of soil moisture between layers and drainage to the
groundwater, sub-surface and groundwater flow and flow through river channels.
For the calculation of potential reference evapotranspiration, potential evaporation from bare soil and open
water, LISFLOOD can be coupled to the LISVAP preprocessing routine (JRC, 2013), especially developed
for this purpose (https://ec-jrc.github.io/lisflood-lisvap/).
In this work LISFLOOD model was applied to the main rivers that cross the selected coastal cities: Frigido
river for Massa (catchment size ~70 km$^2$), Torrent de la Piera and Torrent de San Juan for Villanova (total
size of the two catchments ~40 km$^2$), and Oiartzun river for Oarsoaldea (catchment size ~85 km$^2$). Such
watersheds were represented as gridded domains with 100x100 m cell size (Figure 4a, b, c).
For Frigido river, geomorphological and land cover characteristics were obtained from data available from
Tuscany Region (hydrologically conditioned DEM at 10x10 m resolution, land cover at 1:10000 scale),
while for the other basins data were obtained from EU-DEM v 1.1 25x25 m resolution, Copernicus Land
Monitoring Service (https://land.copernicus.eu) and ISRIC Soil   Grids          250x250           m
(https://www.isric.org).
The meteorological forcing fields extracted from EURO-CORDEX necessary to run the LISFLOOD-
LISVAP models, as reported in section 3, are precipitation (1h), sea level pressure (3h), wind speed (3h),
minimum and maximum air temperature (daily), humidity (daily), shortwave and longwave radiation
(daily).
Output of LISFLOOD are the times series of hourly river discharge in selected points, for each
climatological scenario. Extreme value analysis can then be applied on these long-term time series to obtain
design flood peaks for the selected return periods and the resulting hydrographs to be used as BC for the
hydraulic simulations (whose domains are shown in figure 4d, e, f), as described in Section 5.

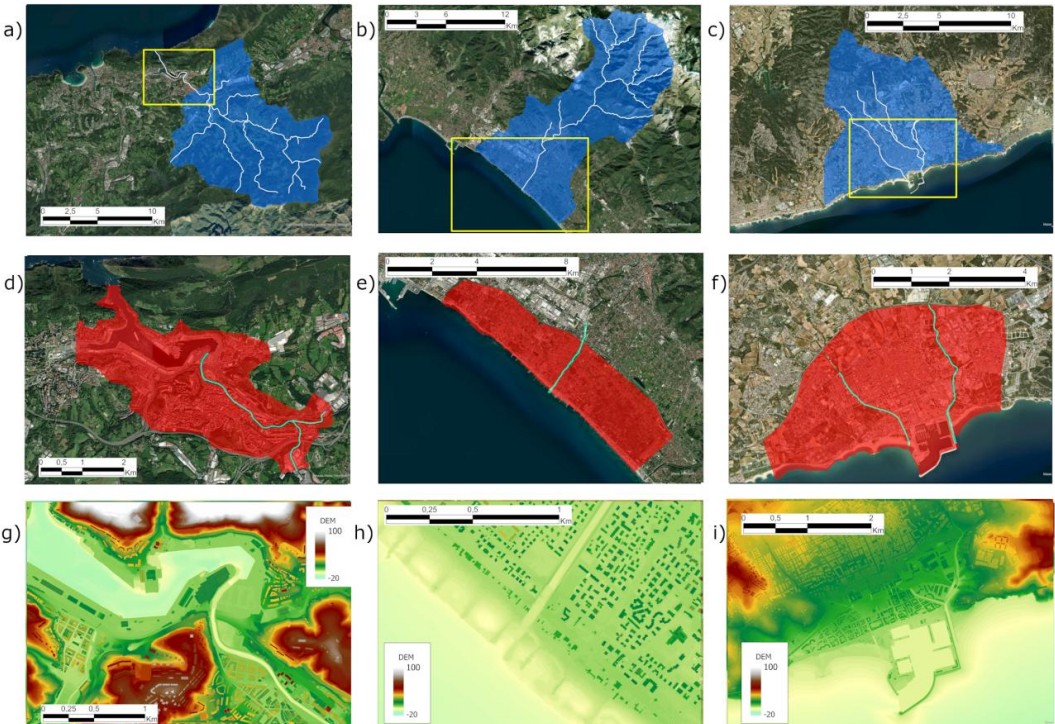

**Fig 4** Top row: View of the domains (blue shading) for the rainfall-runoff hydrological model: a) basin of Oiartzun river, with outlet in Oarsoaldea, b) basin of Frigido river, with outlet in Massa b), basins of Torrent de San Juan and Torrent de la Piera, with outlet in Vilanova c). Middle row: View of the domains of the 2D hydrodynamic modeling (red shading): d) Oarsoaldea, e) Massa, f) Vilanova. Bottom row: Enlargement of the area close to the river mouth, showing the resolution of the employed DEM to create the 2D computational domain: g) Oarsoaldea, h) Massa, i) Vilanova. Base map: Google Satellite imagery (© Google 2024; Imagery © CNES / Airbus, Maxar Technologies, Airbus)

## 5 Modeling floods at urban-scale

In this section, we describe the extreme value analysis to obtain the boundary conditions for the flood simulations at urban scale using the hydrodynamic model implemented at each analyzed city.

### 5.1 Extreme value analysis

The extreme value analysis has been performed for the variables river discharge $Q$ and total water level $\eta_{\text{TOT}}$. Two return period values were determined, 25 and 100 years for each experiment (Historical, RCP4.5, RCP8.5). Furthermore, for the climate projections, two different time windows were analyzed, 2011-2060 and 2051-2100 (Table 1).

In this study, the Generalized Extreme Value (GEV) distribution was employed to model the occurrence of annual maxima values of river discharge $Q$ and total water level $\eta_{\text{TOT}}$, separately. Since our principal



objective is the comparison among different experiments, such a distribution allowed us to be consistent
and to use the same number of events (50) for all the experiments.
The total water level $\eta_{TOT}$ for the Massa and Vilanova cases is equal to the sum of $\eta$ and the runup value,
which is calculated with the Atkinson et al. (2016) equation: $R_{2\%} = 0.92tan\beta\sqrt{H_S L_P} + 0.16H_S$ , where
tanβ is the slope of the beach and $L_P$ is the deep water wavelength at the peak period. Before the calculation
of $R_{2\%}$, the wave height is projected along the orthogonal direction to the coastline, to account for wave
direction. For Oarsoaldea the effect of runup is not included in the computations since we do not simulate
waves within the port.
The Generalized Extreme Value (GEV) distribution can be written as follows (cumulative distribution
function):
$$F(x) = e^{-(1+\xi\frac{x-\mu}{\sigma})^{-\frac{1}{\xi}}}$$

defined for values of $x$ for which $\xi \cdot x > \xi \cdot \mu$-σ. In this equation, $\mu$ is the location parameter, $\xi$ is the shape
parameter, and $\sigma$ is the scale parameter. The shape parameter ξ governs the distribution type: ξ = 0, Type I,
Gumbel distribution; ξ > 0, Type II, Fréchet distribution; ξ < 0, Type III, Weibull distribution (Coles, 2001)
412  .
The parameters $\mu$ , $\sigma$, $\xi$ are estimated from data using the maximum likelihood method. Then, the return
levels $x_{RP}$ for a given return period can be calculated as follows:
$$x_{RP} = \mu + \frac{\sigma}{\xi}((-ln(1 - \frac{1}{RP})^{-\xi}) - 1).$$

To ensure robust estimates of the uncertainties associated with the return levels, the confidence intervals
(CI) at the 95% significance level were calculated using parametric bootstrapping with 500 iterations
(Gilleland, 2020). The statistical analysis has been performed using the R package extRemes: Extreme
Value Analysis (Gilleland and Katz, 2016).
**5.2 Hydrodynamic model**
The effect of extreme storm surge and river flood on the analyzed coastal cities was determined using the
HEC-RAS 6.4 hydrodynamic model (Brunner & US Army Corps of Engineers, 2021). The software couples
the simulation of the flow within a river, solving the one-dimensional Saint-Venant equation, to the two-
dimensional flow on the floodable areas, solving the shallow water equations. Once the water level within
the river bed exceeds the elevation of the levees, water flows on the two-dimensional computational mesh
(the opposite flow is also possible).
The computational domains associated with the three cities are reported in Figure 4d, e, f. Each mesh is
created by overlapping the HEC-RAS computational grid to the digital elevation model (DEM) of the
analyzed area. The system calculates specific elevation-volume relationships for each computational cell,
representing the details of the underlying layer. This allows us to save computational time by setting a lower
resolution for the HEC-RAS mesh with respect to the DEM. For the three cities of Massa, Vilanova and
Oarsoaldea, the DEM is obtained from the LIDAR dataset, at a resolution of 2 m (Figure 4g, h, i). The
HEC-RAS mesh elements have a reference size from 10 to 20 m, except for specific areas (e.g. close to the
coastline, complex urban patterns, etc..) where they are reduced to 5 m. The river geometry is composed
by the river cross sections and additional information of hydraulic structures. For the Massa and Oarsoaldea
cases the geometry comes from a topographic survey, whereas for Vilanova it was extracted from the
LIDAR dataset.




Boundary conditions (BCs) are differently set based on the simulation carried out, as reported in Table 2.
For the river flood simulations the upstream BC is a time series $Q_{RP}(t)$, with peak discharge value $Q_{RP}$ equal
to the return period value. The shape of the hydrograph $Q_{RP}(t)$ is determined as follows: i) the 24 hours
preceding and following the annual maxima are extracted for each year; ii) these 49 hours time series are
superimposed to have maxima in phase and then averaged; iii) the averaged time series is normalized to
obtain $q(t)$, having maximum equal to 1; iv) the $Q_{RP}(t)$ boundary is obtained multiplying $q(t)$ by $Q_{RP}$. Such
a procedure is applied to every run to get the appropriate BC. The downstream BC at the sea is the mean
sea level plus the RSLR, based on the reference scenario $\Delta\eta_{RSLR}$ as reported in Table 2.

|  | River (upstream) BC | Sea (downstream) BC |
|---|---|---|
| River flood | Time series hydrograph $Q_{RP}(t)$ | Mean sea level + $\Delta\eta_{RSLR}$ |
| Coastal flood | Constant hydrograph $Q$ | Time series hydrograph $\eta(t)_{RP}$ + $\Delta\eta_{Tide}$ + $\Delta\eta_{RSLR}$ |

Table 2. Combination of upstream and downstream boundary conditions for the river and coastal flood simulations.

For the coastal flood simulations, considering the inaccuracies inherent in long-term predictions on a
century time scale (Dessay et al., 2009), a statistical approach was preferable to take into account tides and
other factors contributing to the water level of the downstream BC. Specifically, for each site, delta water
levels representing the maximum spring tidal amplitudes $\Delta\eta_{Tide}$ (0.2 m for Massa and for Villanova) and
the predicted sea level rise on a decadal time scale $\Delta\eta_{RSLR}$ (Table 3) were linearly added to the $\eta_{RP}(t)$ time
series to estimate the worst-case scenario for coastal flooding. $\eta_{RP}(t)$ is calculated following the same
procedure employed for the river discharge, with peak value equal to $\eta_{TOT,RP}$. The upstream BC is a constant
value for the river discharge such as the model can run without instabilities and no flood occurs.

| $\Delta\eta_{RSLR}$ | Massa | Villanova | Oarsoaldea |
|---|---|---|---|
| RCP4.5 2011-2060 | 0.150 (-0.036, +0.05) | 0.15 (-0.044, +0.056) | 0.192 (-0.057, +0.061) |
| RCP4.5 2051-2100 | 0.351 (-0.096, 0.131) | 0.349 (-0.105, +0.138) | 0.412 (-0.155, +0.150) |
| RCP8.5 2011-2060 | 0.168 (-0.042, +0.057) | 0.173 (-0.053, +0.062) | 0.229 (-0.079, +0.073) |
| RCP8.5 2051-2100 | 0.464 (-0.137, +0.173) | 0.458 (-0.136, +0.185) | 0.537 (-0.208, +0.203) |

Table 3. Values of RSLR referred to the RCP4.5 and RCP8.5 scenarios, averaged over the reference period, for the analyzed cities.

In Oarsoaldea, we used a slightly different approach for the coastal flood simulations since the tidal
excursion is larger than the extreme return period values: the downstream BC is a semidiurnal tide (up to
2.3 m) added to the $\Delta\eta_{RSLR}$ and to the increase due to the return period value $\Delta\eta_{RP}$.
For the city of Massa a single river called Frigido is simulated and the urban area is divided into two portions
adjacent to the sides of the river (Figure 4d). In Villanova, two river streams are modeled, the easternmost
is the main one, called Torrent de la Piera, whereas the other one (Torrent de Sant Joan) is forced
underground for about 500 meters, just before the rivermouth (Figure 4e). The two-dimensional domain is




split in three subdomains, one between the two rivers and two on their sides. Oiartzun is the main river
modeled for Oarsoaldea, while Lintzirin is its tributary forced underground for most of its length (Figure
4g). In this case the peak discharge of the minor river is scaled in proportion to the basin area (47.4 km$^2$
and 8.7 km$^2$, respectively). The two-dimensional domain is divided into two parts including the Pasaia bay
area.
**6 Results**
**6.1 Extremes for wave climate, water level and river discharge**
A first comparison is performed between the annual maxima of the Historical and Evaluation runs. For the
former, the years from 1973 to 2005 are considered, whereas for the latter those from 1980 to 2012, for an
overall amount of 33 years each. This allows us to have an estimate of the degree of over/under-estimation
we can have on the projections with respect to the actual scenario.

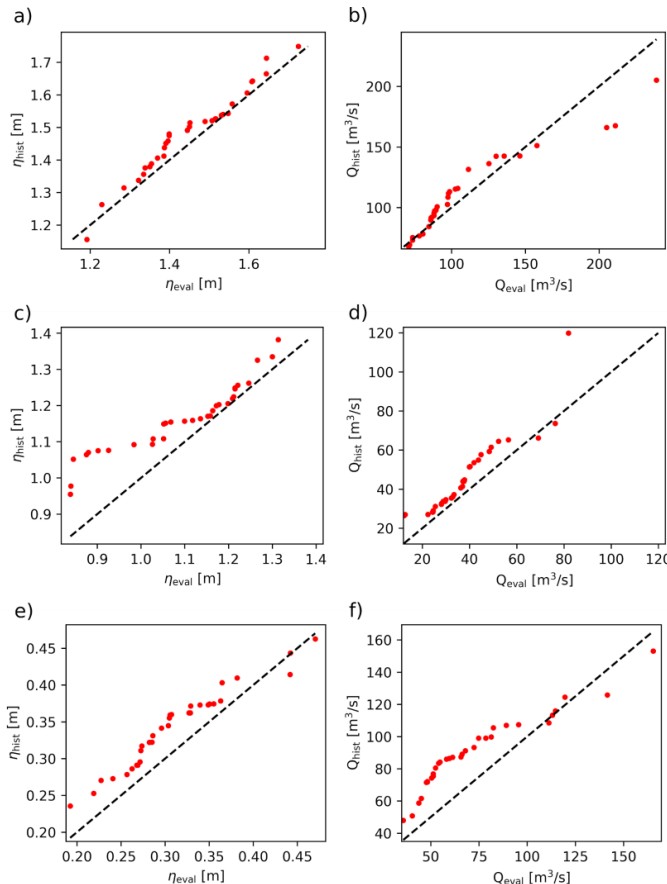

**Fig 5** Quantile-quantile plots between annual maxima from evaluation and historical runs, for the city of Massa a) and
b), Villanova c) and d), and Oarsoaldea d) and e) for the total water level ηTOT (first column), and the peak discharge





Q (second column). For the city of Oarsoaldea η is reported since no runup contribution is considered. Red dots
represent the annual maxima, black dashed line is the 1:1 line
In Figure 5, the quantile-quantile plots for the three analyzed cities for both river discharge and total water
level are reported. Historical and Evaluation annual maxima total water levels in Massa are in agreement
(Figure 5a), whereas Historical river discharge is subject to underestimation only for the highest values
(Figure 5b). Total water levels in Villanova are generally larger for the Historical run with respect to
Evaluation (Figure 5c), whereas river discharge extreme values are correctly estimated except for a single
data (Figure 5d). Oarsoaldea water levels are slightly overestimated by the Historical up to 0.4 m. Also
river discharge values are generally overestimated up to 100 m³/s, then, the largest values tend to be
underestimated (Figure 5f).

| Massa | $\eta_{\text{TOT,RP}}$ [m] | | $Q_{\text{RP}}$ [m³/s] | |
|---|---|---|---|---|
| **Run** | **25 yr (95% CI) [variation to hist %]** | **100 yr (95% CI) [variation to hist %]** | **25 yr (95% CI) [variation to hist %]** | **100 yr (95% CI) [variation to hist %]** |
| **Historical** | 1.736 (-0.062, +0.057) | 1.804 (-0.098, +0.094) | 172 (-31, +42) | 227 (-63, +109) |
| **RCP4.5 2011-2060** | 1.781 (-0.068, +0.041) [+2.6%] | 1.838 (-0.095, +0.057) [+1.9%] | 177 (-34, +53) [+2.9%] | 233 (-72, +155) [+2.6%] |
| **RCP4.5 2051-2100** | 1.770 (-0.076, + 0.084) [+1.95%] | 1.861 (-0.124, + 0.157) [+3.2%] | 210 (-51, +96) [+22.1%] | 307 (-115, +332) [+35.2%] |
| **RCP8.5 2011-2060** | 1.719 (-0.032, +0.013) [-1.0%] | 1.741 (-0.039, 0.014) [-3.5%] | 201 (-36, +52) [+16.9%] | 259 (-74, +136) [+14.1%] |
| **RCP8.5 2051-2100** | 1.739 (-0.050, +0.032) [+0.2%] | 1.783 (-0.066, +0.047) [-1.2%] | 253 (-70, +103) [+47.1%] | 386 (-160 +367) [+70.0%] |

Table 4. Return period values associated with 25 and 100 years for the different runs for the city of Massa for both the total water
level $\eta_{\text{TOT,RP}}$ and the peak discharge $Q_{\text{RP}}$. Numbers in % (in square brackets) represent the variation relative to the historical value.
Table 4, Table 5 and Table 6, show the results of the extreme value analysis of $\eta_{\text{TOT,RP}}$ and $Q_{\text{RP}}$ for the city
of Massa, Villanova and Oarsoaldea, respectively, together with the confidence intervals at 95%
significance level (round brackets) and the percentage increase/decrease (square brackets) with respect to
the Historical values.
For $\eta_{\text{TOT}}$ in Massa, the RCP4.5 scenario shows slightly larger values with respect to the historical run,
whereas the RCP8.5 has similar or slightly lower values. Conversely, extreme $Q$ values tend to grow for
both time windows and further forward in the future for both RCP4.5 and RCP8.5. Nevertheless, the
estimated 100 years peak discharge shows large uncertainty values, especially for the 2051-2100 case for
both RCP4.5 and RCP8.5 runs (Table 4).




| Vilanova | $\eta_{TOT,RP}$ [m] | | $Q_{RP}$ [m³/s] | |
|---|---|---|---|---|
| **Run** | **25 yr (95% CI) [variation to hist %]** | **100 yr (95% CI) [variation to hist %]** | **25 yr (95% CI) [variation to hist %]** | **100 yr (95% CI) [variation to hist %]** |
| **Historical** | 1.360 (-0.055, +0.034) | 1.409 (-0.079, +0.056) | 91 (-20, +27) | 125 (-39, +79) |
| **RCP4.5 2011-2060** | 1.340 (-0.039, +0.022) [-1.5%] | 1.375 (-0.053, + 0.031) [-2.4%] | 87 (-15, +18) [-4.4%] | 107 (-27, +42) [-14.4%] |
| **RCP4.5 2051-2100** | 1.375 (-0.080, +0.058) [+1.1%] | 1.450 (-0.131, +0.097) [+2.9%] | 107 (-22, +25) [+17.6%] | 137 (-41, +59) [+9.6%] |
| **RCP8.5 2011-2060** | 1.401 (-0.075, +0.053) [+3.0%] | 1.473 (-0.108, +0.089) [+4.5%] | 109 (-19, +26) [+19.8%] | 139 (-35, +57) [+11.2%] |
| **RCP8.5 2051-2100** | 1.360 (-0.059, +0.040) [+0%] | 1.411 (-0.081, +0.064) [+0.1%] | 116 (-23, +36) [+27.5%] | 154 (-46, +83) [+23.2%] |

Table 5. Return period values associated with 25 and 100 years for the different runs for the city of Villanova for both the total water level $\eta_{TOT,RP}$ and the peak discharge $Q_{RP}$. Numbers in % (in square brackets) represent the variation relative to the historical value.

Extreme $\eta_{TOT}$ values for the city of Villanova show an increase for the RCP4.5 2051-2100 and for the RCP8.5 2011-2060 scenarios (both 25 and 100 yr RPs), whereas a decrease is found for the RCP4.5 2011-2060. Analogously, $Q$ extreme values are lower than the historical for the RCP4.5 2011-2060 scenario (both 25 and 100 yr RPs), but an increase is observed for all the other cases (Table 5).

| Oarsoaldea | $\eta_{RP}$ [m] | | $Q_{RP}$ [m³/s] | |
|---|---|---|---|---|
| **Run** | **25 yr (95% CI) [variation to hist %]** | **100 yr (95% CI) [variation to hist %]** | **25 yr (95% CI) [variation to hist %]** | **100 yr (95% CI) [variation to hist %]** |
| **Historical** | 0.433 (-0.025, +0.018) | 0.456 (-0.035, +0.025) | 169 (-30, +34) | 209 (-54, +87) |
| **RCP4.5 2011-2060** | 0.444 (-0.039, 0.031) [+2.5%] | 0.486 (-0.061, +0.060) [+6.6%] | 168 (-27, +27) [-0.6%] | 201 (-46, +56) [-3.8%] |
| **RCP4.5 2051-2100** | 0.395 (-0.021, 0.016) [-8.8%] | 0.416 (-0.030, +0.026) [-8.8%] | 176 (-19, +12) [+4.1%] | 195 (-27, +22) [-6.7%] |
| **RCP8.5 2011-2060** | 0.423 (-0.039, +0.035) [-2.3%] | 0.462 (-0.059, +0.077) [+1.7%] | 163 (-18, +15) [-3.5%] | 182 (-27, +32) [-12.9%] |
| **RCP8.5 2051-2100** | 0.416 (-0.022, +0.013) [-3.9%] | 0.436 (-0.030, +0.018) [-4.4%] | 173 (-20, +18) [+2.4%] | 194 (-31, +35) [-7.2%] |

Table 6. Return period values associated with 25 and 100 years for the different runs for the city of Oarsoaldea for both the water level $\eta_{RP}$ and the peak discharge $Q_{RP}$. Numbers in % (in square brackets) represent the variation relative to the historical value.

Concerning extreme water levels in Oarsoaldea, an increase for the RCP4.5 2011-2060 (both 25 and 100 yr RPs) is observed, while all the other cases show decrease or substantial invariance. The extreme river discharge is not subject to significant variations for the 25 years RP, whereas a general slight decrease is observed for the 100 years RP for all scenarios.






## 6.2 Flooded areas

The envelope of the water depth simulated through the hydrodynamic model for coastal and riverine floods
with a 100 yr RP are reported in Figure 6 and Figure 7, respectively.
More specifically, results are reported for the RCP4.5 2051-2100 and RCP8.5 2011-2060 for the three
analyzed cities. The remaining figures of flooded areas, that is: the 100 yr RP coastal floods (Figure S1)
and 100 yr RP riverine floods (Figure S2) cases, and all the 25 y RP cases (Figure S3 for coastal flood and
Figure S4 for riverine flood), are reported in the Supplementary material.
For the city of Massa, the simulations of the future scenarios show an increase in flooded areas, especially
for the RCP4.5 2051-2100 (Figure 6a, d, g). Such a case shows a rise of 60% in flooded volume with respect
to the Historical case, whereas the increase is 7% for the RCP8.5 2011-2060 100 yr RP (Table 7). In general,
coastal flood volume increase in Massa is larger for the furthest time window in the future.
Storm surges in Villanova mainly impact the beach area and the surroundings of the port (Figure 6b, e, h),
and the rise in flooded volume compared to the Historical case is at most 20% (RCP4.5 2051-2100 25 yr
RP, Table 7).

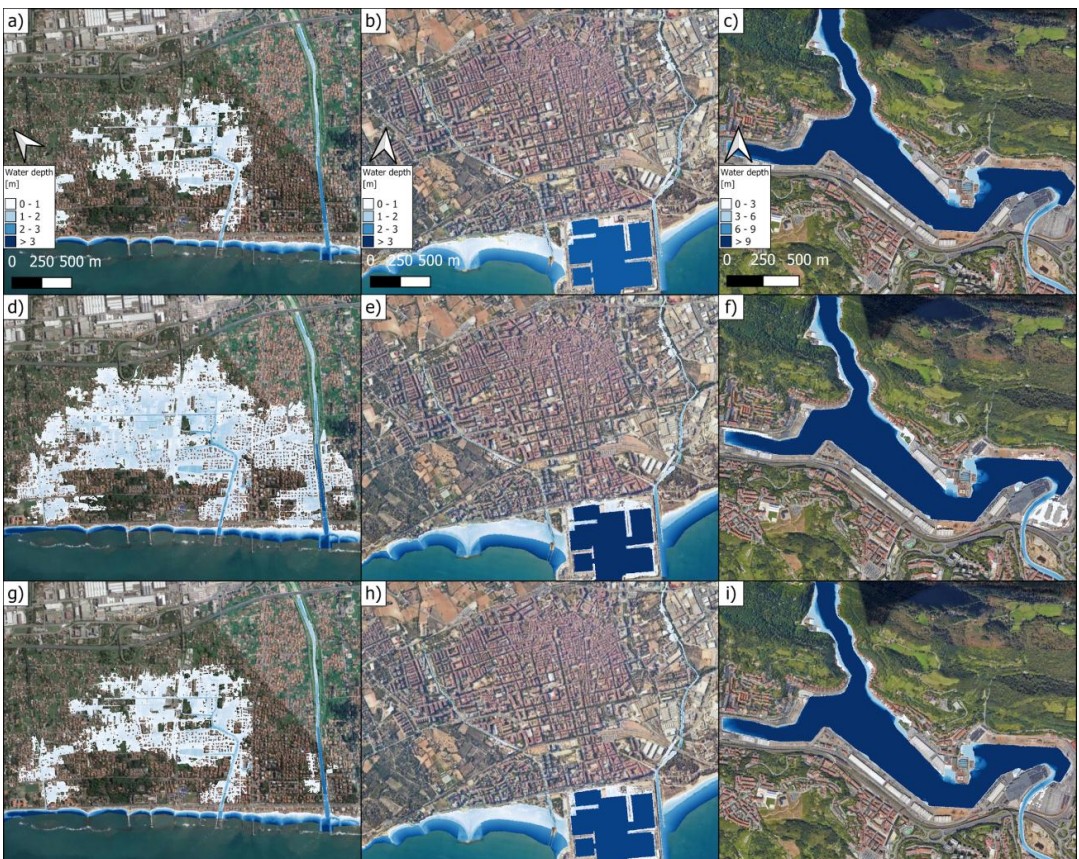

**Fig 6** Hazard maps associated to the 100 years return period coastal flood event for the city of Massa: Historical a),
RCP4.5 2051-2100 d), RCP8.5 2011-2060 g); for the city of Vilanova: Historical b), RCP4.5 2051-2100 e), RCP8.5



2011-2060 h); for the city of Oarsoaldea: Historical c), RCP4.5 2051-2100 f), RCP8.5 2011-2060 i). Base map: Google
Satellite imagery (© Google 2024; Imagery © CNES / Airbus, Maxar Technologies, Airbus)

For Oarsoaldea, the hydrodynamic simulations of coastal flooding do not show substantial variations
between the Historical case and the projections (Figure 6c, f, i). This is confirmed by the flooded volume
variation which is at most 3% for the RCP8.5 2051-2100 100 yr RP (Table 7).

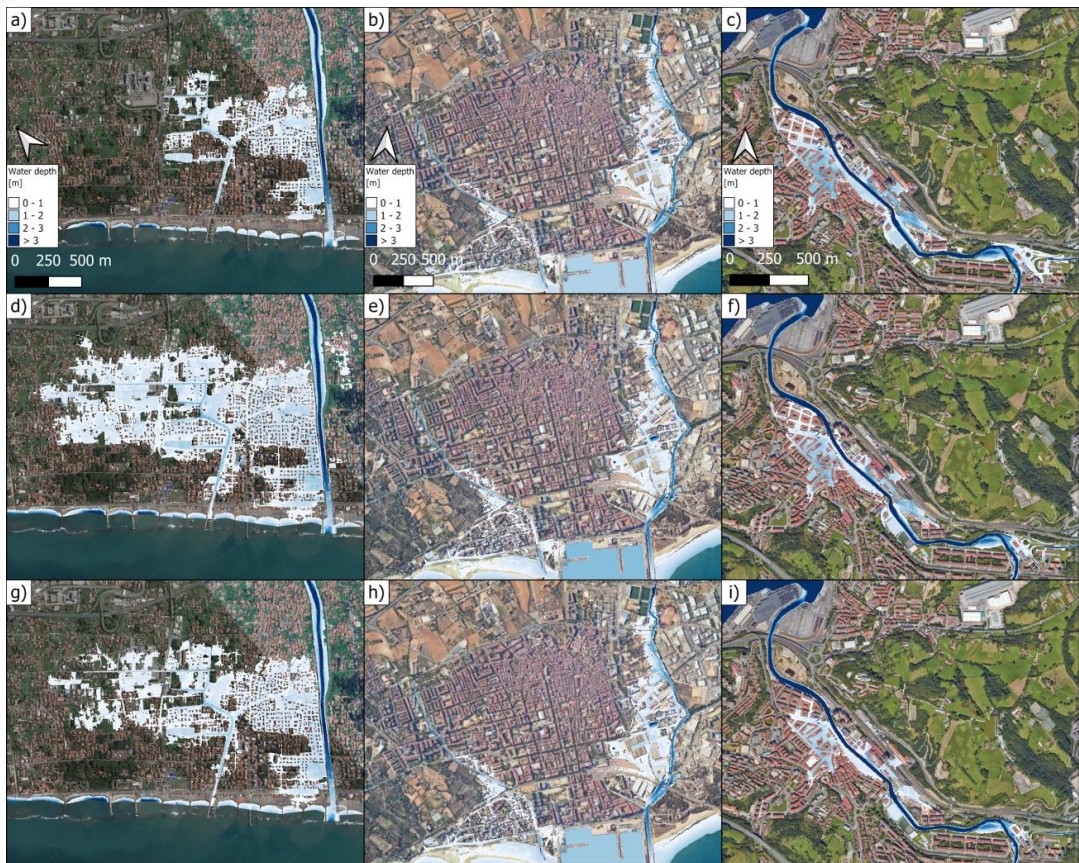

**Fig 7** Hazard maps associated to the 100 years return period riverine flood event for the city of Massa: Historical a),
RCP4.5 2051-2100 d), RCP8.5 2011-2060 g); for the city of Vilanova: Historical b), RCP4.5 2051-2100 e), RCP8.5
2011-2060 h); for the city of Oarsoaldea: Historical c), RCP4.5 2051-2100 f), RCP8.5 2011-2060 i). Base map: Google
Satellite imagery (© Google 2024; Imagery © CNES / Airbus, Maxar Technologies, Airbus)

The results of the 100 yr RP riverine floods hydrodynamic simulations are reported in Figure 7. For the city
of Massa a substantial increase in the flooded area for the RCP4.5 2051-2100 (Figure 7d) and RCP8.5 2011-
2060 (Figure 7g) with respect to Historical case (Figure 7a), is observed. This is consistent with the rise in
flooded volume reported in Table 7, where an increase larger than 200% is seen for both the RPs associated
with the RCP8.5 2051-2100 case.
The visual comparison of Figure 7b, e, h does not allow to clearly detect an increase/decrease in flooded
area with respect to the Historical case for the city of Vilanova. However, the computation of flooded





volume variation shows an increase up to 33% for all cases except for RCP4.5 2011-2060 for both RPs
(Table 7).
Oarsoaldea exhibits a different behavior since the Historical events cause larger floods with respect to most
part of the projections. Even for this city the visual comparison of water depths does not allow us to identify
increase/decrease in flooded areas (Figure 7c, f, i), but the results reported in Table 6 show that rise in
flooded volume around 11% is observed only for the 25 yr RPs for the RCP4.5 for the 2051-2100 time
window. All other cases show a decrease in flooded volume, up to -38% for RCP8.5 2011-2060 100 yr RP.

| Analyzed city | Run | Coastal flood | | Riverine flood | |
|---|---|---|---|---|---|
| | | 25 years | 100 years | 25 years | 100 years |
| Massa | RCP4.5 2011-2060 | +20% | +18% | +7% | +9% |
| | RCP4.5 2051-2100 | +49% | +60% | +84% | +124% |
| | RCP8.5 2011-2060 | +14% | +7% | +51% | +44% |
| | RCP8.5 2051-2100 | +68% | +68% | +218% | +261% |
| Vilanova | RCP4.5 2011-2060 | +1% | +0% | -8% | -20% |
| | RCP4.5 2051-2100 | +8% | +10% | +17% | +11% |
| | RCP8.5 2011-2060 | +6% | +8% | +23% | +15% |
| | RCP8.5 2051-2100 | +9% | +9% | +30% | +33% |
| Oarsoaldea | RCP4.5 2011-2060 | +1% | +1% | -3% | -11% |
| | RCP4.5 2051-2100 | +1% | +1% | +11% | -17% |
| | RCP8.5 2011-2060 | +1% | +1% | -14% | -33% |
| | RCP8.5 2051-2100 | +2% | +3% | +1% | -38% |

Table 7. Percentage change of the flooded volume with respect to the historical run for the three cities of Massa, Vilanova and
Oarsoaldea, for the RCP4.5 and RCP8.5 (2011-2060, 2051-2100) scenarios for both the 25 and 100 years return periods.

**7 Discussion**





Assessing the impacts of future climate scenarios on extreme flood events in coastal cities requires a huge
effort due to the need to integrate processes across multiple scales, from synoptic scale (i.e. storms spanning
~100-1000 km) to local scale. At the urban scale, specific geomorphic features such as landscape elevation
and structural elements can significantly influence flood extent. To address this complexity, we
implemented a multiscale modeling chain tailored for three of the CCLLs under the SCORE Project, but
that can be easily generalized to other coastal cities. We employed unstructured grids modelling approaches
to simulate wave climate (WWIII) and water levels (SHYFEM). These were integrated with the distributed
hydrological model LISFLOOD, and finally coupled within high-resolution urban hydrodynamic
simulations, to capture the interaction between extreme events and urban-specific characteristics, achieving
the spatial granularity needed to capture critical urban-scale flood dynamics. However, this level of detail
comes with a huge computational effort: each of the three models ran simulations equivalent to nearly 300
years, repeated for all analyzed cities.
This consideration was the most significant factor influencing our choice of using a single RCM (and GCM)
rather than a multi-model ensemble approach. In addition, data availability from EURO-CORDEX for all
required variables at a sufficient output frequency and covering the Evaluation, Historical, RCP4.5 and
RCP8.5 runs was ensured only by the ALADIN63 model driven by the ERA-Interim reanalysis and the
CNRM-CM5 GCM. We have given priority to have a continuous dataset at the cost of giving up an
uncertainty estimate based on multi-model ensemble. However, such an estimate, associated with the
extreme values from the analyzed time series, was recovered in the statistical analysis by calculating
confidence intervals through the bootstrap method.
The comparison between the annual maxima from the Evaluation and Historical runs (Figure 5), together
with the information reported in Tables 4, 5 and 6, enables us to assess the reliability of the coastal and
riverine hazard maps (Figures 6 and Figure 7).
Future coastal floods in Massa do not show significant variations in terms of event magnitude compared to
the Historical period. Indeed, the increase/decrease ranges from -3.5% to +3.2 with a predominance of
positive values (Table 3). Considering the 95% CIs, the variability generally lies between -+2.5% and -+5%
of the calculated extreme value for the 2011-2060 and 2051-2100 time windows, respectively. Although an
increase in wave height is projected for theLigurian-Tyrrhenian Sea (De Leo et al., 2024), several factors
may contribute to the observed invariance in total water levels for Massa. The shallow bathymetry in front
of Massa (Figure 3) can act as a sort of filter for the highest offshore waves, leading to a sort of upper limit
for the wave height close to the shoreline which, in turn, affects the total water level through the runup
equation. Additionally, the very high resolution of the modeling near the coast captures local-scale effects
that are often missed by lower-resolution models. Furthermore, the sensitivity of runup to wave height for
Massa's beach slope, combined with the wavelengths associated with the highest waves (70-95 m) is
modest, approximately 0.2-0.25 m. This means that a 1 m increase in wave height produces 0.2-0.25 m
increase in runup. As a consequence, any increase/decrease in wave climate is partially damped. Actually,
the main driver in producing significant differences in flooded volume is the RSLR (Table 3), which causes
the storm surge to penetrate farther inland, resulting in larger flooded volumes (Table 7).
Riverine floods for the RCPs projections in Massa show a substantial increase, even more evident for the
2051-2100 time window. However, this is accompanied by an equal increase in uncertainty. Indeed, the
width of the 95% CI is almost 1.5 times the 100 yr RP for both the RCP4.5 and RCP8.5 2051-2100. Despite
this, the overall increase in extreme $Q_{RP}$ for all the analyzed scenarios and time windows confirms an
increment in future peak river discharges. However, such extremes could be slightly underestimated as
observable from the QQ-plot of Evaluation and Historical annual maxima (Figure 5b). Notwithstanding,




their impact on the ground is further augmented by the increase in relative sea level, whereby the higher
downstream boundary condition hinders the flow toward the sea, resulting in a substantial increase in the
flooded volume (Table 7).
The extension of the flooded area for Vilanova appears not to be affected by storm surges principally due
to the characteristics of the beach zone which is separated from the urban area by a steep positive gradient
in the land elevation which makes the latter higher. A substantial equivalence between the Historical and
the RCP4.5 and RCP8.5 extreme values is observed and the $\Delta\eta_{RSLR}$ ranges between 0.15 and 0.458 m
(Table 3). Even though the increase in flooded volume is always positive (Table 7), the flooded area is not
enlarged (Figure 6b, e, h) and the only area which is interested in an enlargement of the flooded surface is
the one adjacent to the port.
The riverine floods associated with projections are generally characterized by an increase in flooded volume
with respect to the Historical (from +11% to +33%), but for the RCP4.5 2011-2060 25 and 100 yr RPs (-
8% and -20%), as reported in Table 7. Concerning the $Q_{RP}$ values, the higher the extreme value, the larger
the CI width. However, a substantial increase in river discharge is observable, in agreement with the flooded
volume. The comparison of annual maxima from Evaluation and Historical (Figure 5d) suggests no
underestimation/overestimation, even if the largest value could lead one to think of an overestimation. The
additional increase in flooded volume (Table 7) compared to the maxima in river discharge (Table 5) is
primarily attributed to the RSLR, similar to the findings for Massa.
For the city of Oarsoaldea the port area has been designed to face tidal excursions around 2 m. The extreme
values associated with both 25 and 100 yr RP range between 0.395 and 0.486 m. Table 5 reports increases
(RCP4.5 2011-2060) and decreases (RCP4.5 2051-2100 and RCP8.5 2051-2100) of the extreme water level
for the projections compared to the Historical, consistent with the findings of Vousdoukas et al. (2017).
The modest rise in flooded volume (+1% to +3%, Table 7) is mainly attributable to the RSLR.
For river discharge, a generalized decrease in peak $Q_{RP}$ values is observed, with the width of the 95% CI of
the same order of magnitude of the variation with respect to the Historical period, and an expected slight
underestimation of the projected extremes (Figure 5f).
The use of annual maxima to perform the EVA has the disadvantage of eliminating a lot of significant data.
To make greater use of the time series produced, we performed two additional analyses for the city of Massa
for both $\eta_{TOT}(t)$ and $Q(t)$. We calculated the cumulative time a variable persists over a fixed threshold, that
is chosen as the 99.5%-ile and the 99.9%-ile of the Historical period time series for the total water level and
river discharge, respectively (Figure 8a and 9a). Furthermore, we determined the number of events per year
higher than specific values, clustering the events by decade (Figure 8b and 9b).
Figure 8a shows that the increase in RSL is the main driver for the $\eta_{TOT}$ increase for the cumulative time
above a certain high level. It is also confirmed by Figure 8b where a trend in the increase of extreme events,
without the effect of RSLR, is not clearly observable.



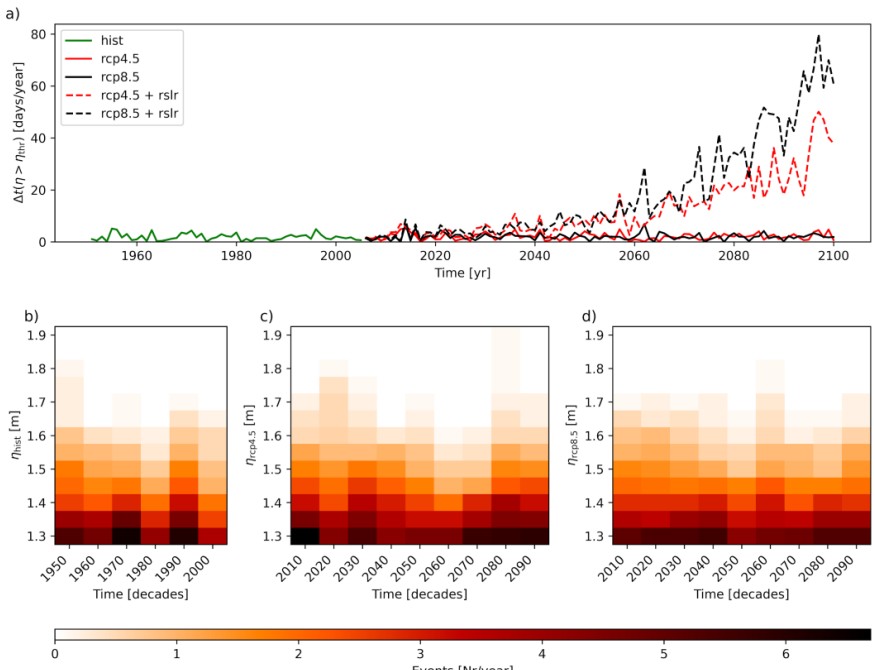

**Fig 8** a) Cumulative duration of total water level above the 99.5 %-ile in days per year for HIST (green line), RCP4.5
(red line), RCP8.5 (black line), RCP4.5 and RCP8.5 plus the effect of RSLR (red dashed line and black dashed line,
respectively), for the city of Massa. Number of events per year with peak values larger than specific values, grouped
by decades for: HIST b), RCP4.5 c) and RCP8.5 d)
Concerning river discharge, a slight positive trend for the cumulative time $Q(t)$ persists above the 99.9%-
ile Historical value, is detectable (Figure 9a). Moreover, an increase in the number of extreme events is
observed, especially for the RCP8.5 scenario, even if the obtained patch is quite noisy. This can be ascribed
to the fact that we used only one RCM.

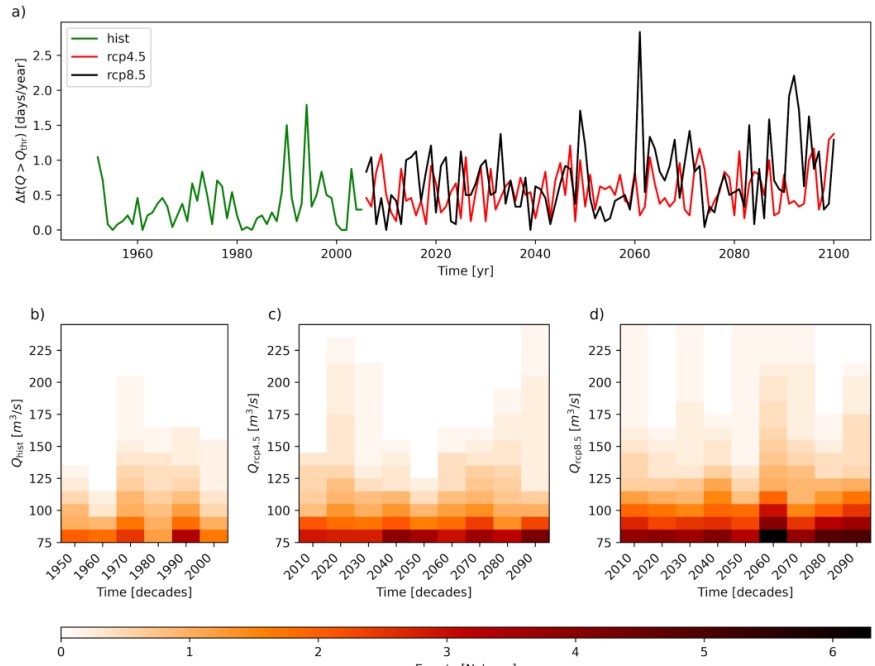

**Fig 9** a) Cumulative duration of water discharge above the 99.9 %-ile in days per year for HIST (green line), RCP4.5 (red line), RCP8.5 (black line), RCP4.5 and RCP8.5 plus the effect of RSLR (red dashed line and black dashed line, respectively), for the city of Massa. Number of events per year with peak values larger than specific values, grouped by decades for: HIST b), RCP4.5 c) and RCP8.5 d)

A potential limitation concerning the analysis of extreme events is related to compound events (Ghanbari et al., 2020; Gori & Lin, 2022). In this work we consider non-interacting storm surges and river discharges. Such a choice is aimed at simplifying the approach and having greater control on each driver of a flood event. Furthermore, the extreme value analysis of compound events lead to some difficulties and approximations related to the identification of a "compound event". In general, focusing only on two variables, we look for large values, in one or both variables, which are temporally distant less than a specific threshold. The method to identify "compound events" varies on the basis of the different studies and scales. This aspect, together with the choice of the couple of values associated with a specific RP curve, tends to enhance the complexity and the degrees of freedom of the problem. Considering the present work as an introductory paper describing the whole modeling chain and its applications, and given the availability of continuous time series, we intend to pursue future research by focusing directly on impacts. That is, we intend to run the hydrodynamic model using as BCs the whole time series (excluding the periods where both $Q$ and $\eta_{TOT}$ are low), and analyze the statistical properties of the water depth as a consequence of flood events. In such a way it is possible to by-pass all the issues related to the definition and identification of compound events. Nevertheless, the availability of these long term simulated discharge time series can also be a valuable dataset for further analysis on hydrological regimes e.g. droughts.

It is important to stress that the errors which accumulate along the modeling chains are difficult to estimate and are the results of unavoidable approximations. Furthermore, we are running hydrodynamic simulations





where the environment (e.g. buildings, structure, etc…) do not change in time, which is an unlikely
circumstance. As a consequence, obtained results have to be considered as an indication of a trend rather
than a solid prediction of the future.
On the one hand, we are making a strong assumption, considering the surrounding environment does not
change over time. On the other hand, the knowledge of specific characteristics of the analyzed area are
crucial in modeling the impact of flood events. A coarse starting DEM of around 20 m resolution cannot
even resolve streets and spaces between buildings, potentially blocking the flow and significantly changing
the flooding pattern. These are aspects that have to be taken into account when evaluating the obtained
results associated with uncertain future scenarios.
**8 Conclusion and outlook**
In this work we present a modeling chain to transfer synoptic scale atmospheric information to the scale of
coastal cities with the goal of estimating changes in the impact of extreme riverine and coastal flood events
- specifically in terms of flooded area and volume - under the RCP4.5 and RCP8.5 climate change scenarios,
compared to Historical conditions. We use atmospheric data from the ALADIN63 RCM from the EURO-
CORDEX dataset to drive three numerical models: WWIII for wave climate, SHYFEM for water levels,
and LISFLOOD for river discharge. Model outputs are then processed to generate synthetic extreme events,
which are then used to simulate coastal and riverine floods through a high-resolution hydrodynamic model
(HEC-RAS). This model is specifically implemented for the domains of three coastal cities selected within
the SCORE Project: Massa (Italy), Vilanova i la Geltrù, and Oarsoaldea (Spain). Wave climate data are
further used to calculate wave runup, which is combined with water levels to determine total water levels
$\eta_{TOT}$.
The extreme value analysis of total water levels $\eta_{TOT}$ and river discharge $Q$ reveals both increase and
decrease in RCP4.5 and RCP8.5 extremes compared to Historical extremes, depending on the different
locations, with larger uncertainties associated with high extreme values and longer-term projections (2051-
2100). The increase/decrease in flooded volume is not necessarily related to increase/decrease in extremes
but it depends by relative sea-level rise RSLR and to specific local features of each coastal city.
Massa is particularly vulnerable to RSLR, which facilitates the inland propagation of coastal floods,
increasing the water volume up to 68%. Additionally, RSLR hinders river flow into the sea, exacerbating
riverine floods and potentially doubling water volume. This is further compounded by an increase in future
extreme river discharge (ranging from +2.9% to +70%), especially under the RCP8.5 scenario. In contrast,
Vilanova i la Geltrù is not significantly affected by storm surges due to its geomorphic structure, whereas
the riverine extreme floods tend to generally increase in the future according to RCP4.5 and RCP8.5 (up to
+27.5% for peak river discharge and +33% for water volume). Oarsoaldea, on the other hand, is well
protected against storm surges and the flood extension appears to be relatively insensitive to the differences
between Historical, RCP4.5 and RCP8.5 scenarios. Riverine floods in Oarsoaldea show a decrease in extent
for the 100 yr RP but slightly increase for the 25 yr RP in the 2051-2100 timeplay. These results reflect the
complex interplay between extreme events and RSLR.
This study highlights the importance of employing high resolution modeling, as local characteristics
significantly influence flood impacts and the analysis of the effects of future extreme events.
Future developments include the use of long-term time series of $\eta_{TOT}$ and Q to continuously force the
hydrodynamic model, excluding periods associated with low values. This impact-based approach could
eliminate the need for EVA for different events, including compound events and enables a direct analysis



of their interaction on the ground, providing a statistical assessment of water depth, flood extent and water
volume time series.

**Author contributions**

**B.M.**: conceptualization, formal analysis, investigation, methodology, software, visualization, writing - original draft, writing - review and editing. **C.F.**: conceptualization, investigation, methodology, writing - original draft. **C.A.**: conceptualization, investigation, methodology, visualization, writing - original draft. **T.S.**: conceptualization, investigation, methodology, writing - original draft. **A.I.**: formal analysis, project administration, investigation. **P.R.**: writing - original draft, writing - review & editing. **M.R.**: investigation, visualization, writing - review & editing. **P.M.**: data curation, investigation, visualization. **S.M.**: formal analysis, investigation, writing - review & editing. **V.G.**: data curation. **O.A.**: conceptualization, funding acquisition, project administration, writing - review and editing. **C.M.**: project administration. **G.S.**: funding acquisition, project administration, writing - review and editing. **B.C.**: conceptualization, supervision, funding acquisition, project administration, writing - review and editing.

**Competing interests**

The authors declare that they have no conflict of interest.

**Funding**

This research was supported by the project SCORE (Smart Control of the Climate Resilience in European Coastal Cities), funded by the European Commission's Horizon 2020 research and innovation programme under grant agreement No. 101003534.

**Acknowledgments**

Thanks also to the European Union—NextGenerationEU and the Ministry of University and Research (MUR), National Recovery and Resilience Plan (NRRP), Mission 4, Component 2, Investment 1.5, project "RAISE—Robotics and AI for Socio-economic Empowerment" (ECS00000035); and the EU - Next Generation EU Mission 4 "Education and Research" - Component 2: "From research to business" - Investment 3.1: "Fund for the realisation of an integrated system of research and innovation infrastructures" - Project IR0000032 – ITINERIS - Italian Integrated Environmental Research Infrastructures System.

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
