# Peer review of "Multiscale Modeling for Coastal Cities: Addressing Climate Change Impacts on Flood Events at Urban-Scale"

_EGUsphere, 2025_

## Referee Comment (RC1)

**Review of the paper "Multiscale Modeling for Coastal Cities: Addressing Climate Change Impacts on Flood Events at Urban-Scale" submitted by Michele Bendoni**

This study introduces a comprehensive modeling framework to assess the impacts of climate change on coastal and riverine flooding in European coastal cities. By integrating atmospheric data from the ALADIN63 regional climate model under the EURO-CORDEX project, researchers simulate future flood scenarios under RCP4.5 and RCP8.5 climate pathways. The modeling chain employs a wave model, storm surge model, river discharge model based on rainfall-runoff processes, and HEC-RAS for hydrodynamic simulations to assess flood extents and depths. The findings underscore the importance of high-resolution, integrated modeling approaches to accurately project and manage future flood risks in coastal urban areas.

I would like to compliment the authors for a well-written manuscript. The paper introduces a framework for assessing coastal and inland flooding at the urban scale in the European region under climate change. I recommend publication of the manuscript after properly addressing the following comments.

Main comments:

1. The manuscript does not provide any information on the validation of the numerical models used in the framework, specifically the wave model, storm surge model, river discharge model, or the hydrodynamic model. While the paper presents a generalized workflow, the quantitative conclusions (e.g., flood volumes, durations, extents) rely heavily on the reliability of the model outputs. Without presenting model validation or calibration results, it is difficult to assess the robustness of the findings. This is particularly critical for extreme hydrologic/hydrodynamic modeling, where prior studies have shown that numerical models often fail to estimate peak magnitudes accurately unless properly calibrated. I strongly recommend including a validation section for each model, either in the main text or supplementary materials, so that readers can better understand the limitations and reliability of the framework and derived results.

2. To provide boundary conditions for the hydrodynamic model, the time series of Q and total water levels are obtained by averaging superimposed 49-hour time series around the annual maxima peaks. The time series surrounding the annual maxima may exhibit multiple peaks and varied shapes across a wide range, particularly for Q, depending on the upstream catchment and rainfall characteristics. Therefore, averaging at each timestep and treating it as the representative hydrograph may overlook temporal dependencies and could be unrealistic. A similar issue may arise for storm surge and/or total water levels (see Quinn et al., 2014; Santamaria-Aguilar et al., 2017). To support the approach used in this study, a figure showing all individual and averaged hydrographs (possibly in the supplementary material) or appropriate citations of relevant studies that have previously applied this method would strengthen the justification.

3. The purpose of comparing the historical and evaluation simulations is unclear. These runs differ in only 7 years of annual maxima, with the remaining 26 years overlapping. It's mentioned that this analysis is done to "test the ability of the model to reproduce observable extreme events." I would like to see some discussion on how this is achieved in the author's reply in the open discussion. Additionally, while the authors suggest that this comparison of quantile plots helps to estimate

the degree of over- or under-estimation in projections (e.g., L 624-L625), the interpretation of these results and their contribution to the main analysis are not well explained.

4.

Other Comments:

1.  Figures 6 & 7: The term "envelope of the water depth" is unclear. Do the flood maps represent the maximum depth reached at each grid cell throughout the simulation period, or are they snapshots at a specific moment in time? Please clarify this in the text or figure captions.

2.  The authors claim that the uncertainty, which was not accounted for due to the absence of a multi-model ensemble, was recovered in the statistical analysis by calculating confidence intervals through the bootstrap method. However, these represent fundamentally different types of uncertainty and should not be considered as "recovered" from one another.

3.  Line 87: Typo: the word "and" is misspelled.

4.  Line 343: The sentence implies that the shape, timing, and magnitude of streamflow are determined solely by upstream hydrological processes. However, downstream conditions such as tides and storm surge can also play a significant role. Please revise the sentence to reflect this more accurately.

5.  Lines 614–616: The phrase "associated with the highest waves" is unclear whether it refers to wave heights or wavelengths. Please specify.

6.  Lines 654–657: It is unclear why the analysis in this section was conducted only for one location instead of all three sites. The phrase "number of events per year higher than specific values" would benefit from clarification; please indicate what those specific threshold values are.

Quinn, N., Lewis, M., Wadey, M. P., and Haigh, I. D.: Assessing the temporal variability in extreme storm-tide time series for coastal flood risk assessment, J Geophys Res Oceans, 119, 4983–4998, https://doi.org/10.1002/2014JC010197, 2014.

Santamaria-Aguilar, S., Schuerch, M., Vafeidis, A. T., and Carretero, S. C.: Long-term trends and variability of water levels and tides in Buenos Aires and Mar del Plata, Argentina, Front. Mar. Sci., 4, 1–15, https://doi.org/10.3389/fmars.2017.00380, 2017.

---

## Author Response (AR1)

We thank the referee for the useful and productive comments. Here, we reply to the raised issues. Referee comments are reported in blue italic font.

**Referee 1**

**Main comments**

1. *The manuscript does not provide any information on the validation of the numerical models used in the framework, specifically the wave model, storm surge model, river discharge model, or the hydrodynamic model. While the paper presents a generalized workflow, the quantitative conclusions (e.g., flood volumes, durations, extents) rely heavily on the reliability of the model outputs. Without presenting model validation or calibration results, it is difficult to assess the robustness of the findings. This is particularly critical for extreme hydrologic/hydrodynamic modeling, where prior studies have shown that numerical models often fail to estimate peak magnitudes accurately unless properly calibrated. I strongly recommend including a validation section for each model, either in the main text or supplementary materials, so that readers can better understand the limitations and reliability of the framework and derived results.*

We thank the reviewer for this important and constructive comment, which helped us improve the transparency and completeness of our modeling approach. In response, we have added a dedicated section to the Supplementary Material detailing the calibration and validation procedures adopted for the water level, wave climate, and hydrological models. While we could not find suitable observational data to directly validate the hydrodynamic model (HEC-RAS) in reproducing flooding events in the study areas, we acknowledge this limitation clearly in the revised text.

Nevertheless, the use of HEC-RAS in flood modeling is widely established in the literature, including applications to coastal and compound flooding scenarios. Its reliability has been demonstrated in several studies, including urban-scale and coupled modeling frameworks (e.g., Pandey et al., 2021; Bennett et al., 203), which supports its adoption in our case.

To enhance clarity, we have uploaded the cal_val.pdf document containing the newly added section to the Supplementary Material.

We also revised the main text (lines 261–263) as follows:

"*In this section, we describe the implementation of the three numerical models: WaveWatch III, SHYFEM and LISFLOOD, employed to perform the main part of the downscaling procedure. Each of the models has a particular setup on the basis of the analyzed coastal city. Furthermore, a calibration/validation procedure has been carried out for each of them to have an estimate of their skill to reproduce observed events. The detailed description of the different procedures is reported in the Supplementary Material.*"

For the sake of clarity, we use the expression "calibration/validation" in a broad sense, not strictly referring to the use of one dataset to tune the model parameters (calibration), and a separate

dataset to assess model skill (validation). Due to limitations in data availability, we show that the models are nonetheless able to reproduce observed quantities under different conditions: through a systematic trial-and-error calibration (for the water level model SHYFEM), by adopting a previously validated setup (in the case of the wave climate model WWIII), or by relying on community-driven configurations (as for the hydrological model LISFLOOD). This clarification is also included in the cal_val.pdf document provided in the Supplementary Material.

Additionally, we plan to revise the Results section (line 497) to include one key outcome of the calibration procedure:

"*As a result of the calibration and validation process, we observed a tendency of the evaluation run to underestimate the most extreme values measured by wave buoys (see Supplementary Material, Figure S12).*"

2. *To provide boundary conditions for the hydrodynamic model, the time series of Q and total water levels are obtained by averaging superimposed 49-hour time series around the annual maxima peaks. The time series surrounding the annual maxima may exhibit multiple peaks and varied shapes across a wide range, particularly for Q, depending on the upstream catchment and rainfall characteristics. Therefore, averaging at each timestep and treating it as the representative hydrograph may overlook temporal dependencies and could be unrealistic. A similar issue may arise for storm surge and/or total water levels (see Quinn et al., 2014; Santamaria-Aguilar et al., 2017). To support the approach used in this study, a figure showing all individual and averaged hydrographs (possibly in the supplementary material) or appropriate citations of relevant studies that have previously applied this method would strengthen the justification.*

We thank the reviewer for the valuable comment, which helped us improve the clarity and communication of the results. In response, we plan to add in the Supplementary material three additional figures, one for each city, depicting the time series of discharge Q and total water level for the Historical, RCP45 and RCP85 runs. Each subplot contains the superimposition of the time series around the yearly maximum (grey lines), the average (black line) and a red shaded area representing +/- one standard deviation.

[Figure]

Fig S5 Shape of the synthetic hydrograph for the city of Massa. Centered yearly maximum time series are reported in grey, average in black, and +/- one standard deviation is reported as shaded area. a), b), c) refer to discharge for Historical, RCP45 and RCP85 runs, respectively. d), e), f) refer to total water level for Historical, RCP45 and RCP85 runs, respectively.

[Figure]

Fig S6 Shape of the synthetic hydrograph for the city of Villanova. Centered yearly maximum time series are reported in grey, average in black, and +/- one standard deviation is reported as shaded area. a), b), c) refer to discharge for Historical, RCP45 and RCP85 runs, respectively. d), e), f) refer to total water level for Historical, RCP45 and RCP85 runs, respectively.

[Figure]

Fig S7 Shape of the synthetic hydrograph for the city of Oarsoaldea. Centered yearly maximum time series are reported in grey, average in black, and +/- one standard deviation is reported as shaded area. a), b), c) refer to discharge for Historical, RCP45 and RCP85 runs, respectively. d), e), f) refer to water level for Historical, RCP45 and RCP85 runs, respectively.

The reported figures show different variability ranges (red shaded area), depending on the city and the variable analyzed. Nonetheless, the spread among realizations remains moderate and does not compromise the representativeness of the synthetic hydrograph. Quinn et al. (2014) investigated the temporal variability of storm tides by aligning observed peaks and explicitly analyzing tide-surge interactions. In our study, we opted for a simpler and more precautionary approach: we phase-aligned only the residual components, assuming that the tidal level coincides with its peak during the extreme event.

Figures S5, S6, and S7 demonstrate that the variability of residuals is generally stable throughout the event duration. While this represents a worst-case scenario, it still allows for consistent comparisons across climate scenarios—particularly since climate change primarily affects residual components and mean sea level, rather than tidal dynamics.

Although our method does not capture the probabilistic variability arising from tide–surge interactions, it provides a robust and comparable framework across cities and scenarios. Moreover, direct coupling of projected tidal time series with predicted residuals would likely introduce further uncertainty, given the long timescales and limitations in forecasting tidal constituent changes. As Quinn et al. (2014) noted, this added variability would make it more difficult to clearly define floodable areas.

Therefore, while our method may appear simplified, it is appropriate and justified for the purpose and temporal scale of this study.

We acknowledge that trends in tidal constituents, as highlighted by Santamaria-Aguilar et al. (2017), were not explicitly included in this study. These long-term (century-scale) changes in tidal amplitudes are driven by meteorological, oceanographic, and hydrographic variability, and are known to be highly site-specific. Since such variations have been detected from past observations rather than derived from predictive models, and no consolidated numerical methods are currently available to project them, we consider it reasonable not to include them directly. Nevertheless, we mention this limitation in Section 5 and in the Discussion as a relevant aspect deserving further investigation.

We are aware that averaging extreme events around their peaks smooths out their temporal variability. Nevertheless, our intention was to derive a representative event shape that captures the typical rise and fall of the signal. The limitations associated with this approach, such as the possible omission of temporal dependencies and multi-peak structures, are unavoidable when only the maximum value (linked to a return period) is available for constructing synthetic hydrographs. To overcome this, an impact-based approach can be adopted. In such a framework, the statistical analysis is conducted *a posteriori*, based on simulated water depths resulting from multiple realizations of extreme events, using full time series of Q and total water level as boundary conditions. This methodology, already mentioned at lines 687–703, allows a more explicit consideration of variability in time.

To our knowledge, the derivation of a synthetic hydrograph starting from the maximum discharge was proposed by Brunner et al. (2017). Here we do not refer to methods where the input is rainfall with a given return period and duration, subsequently transformed into discharge using a rainfall-runoff model. In their work, Brunner et al. present a detailed procedure to construct synthetic design hydrographs (SDHs) based on "the fitting of probability density functions to observed flood hydrographs of a certain flood type taking into account the dependence between the design variables peak discharge and flood volume". Their method also involves a normalization step, similar to what we adopted. However, we intentionally opted for a simpler procedure that can also be extended to total water levels, a case for which -to our knowledge- no comparable methodology is available.

We expanded the discussion accordingly, addressing this point and clarifying our methodological choices with reference to the available literature.

We plan to add the following from lines 444-445 onwards:
"*Such a procedure is applied to every run to get the appropriate BC. The figures showing the superimposition of the annual maxima events for river discharge and water level for the three cities of Massa, Villanova and Oarsoladea, are reported in the Supplementary Material.*"

In addition, we plan to add the following at line 457:
"*This approach does not take into account long-term trends potentially present in the tidal constituents as observed by Santamaria-Aguilar et al. (2017).*"

We finally plan to expand the discussion after line 694.
"*Additionally, another issue that can be overcome in case the impact-based approach is employed, is that related to the creation of adequate synthetic boundary conditions associated with specific return periods. The choice to average the extreme events superimposed in phase at*

*the peak can smooth out their variability, given that they show different variability ranges based on the selected city and variable (Quinn et al., 2014). Nevertheless, the obtained variability ranges do not exceed the magnitude of the associated extreme event (see Figure S5, S6, S7 in the Supplementary Material). It was our intent to derive a shape for the time series that is representative of the main behaviour of the analyzed variable during its rise and fall around the maximum. The derivation of a synthetic hydrograph starting from the maximum discharge is proposed by Brunner et al., (2017), where they retrieve a synthetic design hydrograph based on "the fitting of probability density functions to observed flood hydrographs of a certain flood type taking into account the dependence between the design variables peak discharge and flood volume". They also pass through a normalization step, similar to what we carried out. However, we tried to keep a simpler approach that can be also extended to the total water levels, for which we did not find an analogous procedure.*

*Accounting for the tide by adding a fixed $\Delta\eta_{\text{Tide}}$ to the extreme event hydrograph (Massa and Villanova), or by simulating a semidiurnal tide (Oarsoaldea) as a boundary condition can overlook the long-term (century-scale) modifications in tidal ranges. Santamaria et al. (2017), using site specific past observations, found they are driven by meteorological, oceanographic, and hydrographic variability. The difficulty to forecast them using numerical tools partly justifies the decision not to explicitly include this aspect in the present study.*"

Added references

Brunner, M. I., Viviroli, D., Sikorska, A. E., Vannier, O., Favre, A. C., & Seibert, J. (2017). Flood type specific construction of synthetic design hydrographs. Water Resources Research, 53(2), 1390-1406.

Quinn, N., Lewis, M., Wadey, M. P., & Haigh, I. D. (2014). Assessing the temporal variability in extreme storm-tide time series for coastal flood risk assessment. Journal of Geophysical Research: Oceans, 119(8), 4983-4998.

Santamaria-Aguilar, S., Schuerch, M., Vafeidis, A. T., & Carretero, S. C. (2017). Long-term trends and variability of water levels and tides in Buenos Aires and Mar del Plata, Argentina. Frontiers in Marine Science, 4, 380.

*3. The purpose of comparing the historical and evaluation simulations is unclear. These runs differ in only 7 years of annual maxima, with the remaining 26 years overlapping. It's mentioned that this analysis is done to "test the ability of the model to reproduce observable extreme events." I would like to see some discussion on how this is achieved in the author's reply in the open discussion. Additionally, while the authors suggest that this comparison of quantile plots helps to estimate the degree of over- or under-estimation in projections (e.g., L 624-L625), the interpretation of these results and their contribution to the main analysis are not well explained.*

We start from the assumption that the Evaluation run, being a reanalysis, closely reflects observed reality as it benefits from data assimilation. In contrast, the Historical run is a free simulation constrained only by the radiative forcing.

To assess the model's ability to reproduce observable extreme events, we compare the statistical distribution of annual maxima from the two runs using quantile–quantile (QQ) plots. If the values align along the 1:1 line, this suggests that the Historical run can statistically reproduce the distribution of observed extremes. Deviations from the 1:1 line indicate systematic under- or over-estimation. Since the RCP45 and RCP85 scenarios are based on the same model, we extend this diagnostic to those projections, assuming that similar biases may occur.

We plan to add the following after lines 624-625:
"*Indeed, we make the hypothesis that the Evaluation run, being a reanalysis, is close to reality given that It benefits from a data assimilation procedure, thus incorporating the information from observations.*"

**Other comments**

*1. Figures 6 & 7: The term "envelope of the water depth" is unclear. Do the flood maps represent the maximum depth reached at each grid cell throughout the simulation period, or are they snapshots at a specific moment in time? Please clarify this in the text or figure captions.*
The flood maps represent the maximum water depth reached at each cell during a simulated event. The following sentence (line 531):
"*The envelope of the water depth simulated through the hydrodynamic model for coastal and riverine floods with a 100 yr RP are reported in Figure 6 and Figure 7, respectively.*"
will be modified to:
"*The envelope of the water depth, that is the spatial distribution of the maximum water depth reached at each computational cell during the hydrodynamic simulation, are reported in Figure 6 and Figure 7 for coastal and riverine floods with a 100 yr RP, respectively.*"

*2. The authors claim that the uncertainty, which was not accounted for due to the absence of a multimodel ensemble, was recovered in the statistical analysis by calculating confidence intervals through the bootstrap method. However, these represent fundamentally different types of uncertainty and should not be considered as "recovered" from one another.*
We changed the text (lines 599-601):

"*However, such an estimate, associated with the extreme values from the analyzed time series, was recovered in the statistical analysis by calculating confidence intervals through the bootstrap method.*"
As                                                                                                    follows:
"*We tried to partially compensate for the lack of such an uncertainty estimation, by calculating confidence intervals through the bootstrap method in the statistical analysis, although this is a different source of uncertainty.*"

*3. Line 87: Typo: the word "and" is misspelled.*
Corrected.

*4. Line 343: The sentence implies that the shape, timing, and magnitude of streamflow are determined solely by upstream hydrological processes. However, downstream conditions such as tides and storm surge can also play a significant role. Please revise the sentence to reflect this more accurately.*

The sentence at lines 342-343:

"*In order to model river floods, it is necessary to define the discharge hydrographs, i.e. the evolution in time of flow rate in given cross sections.*"

Is modified as follows:

"*In order to model river floods, it is necessary to define the inflow discharge hydrograph as a boundary condition, i.e. the evolution in time of flow rate in the upstream cross section.*"

We also added the following proposition after line 348:

"*Moreover, the downstream boundary condition defined by the water level at the outlet affects the evolution of the hydrograph while traveling along the river (see Section 5.2).*"

*5. Lines 614–616: The phrase "associated with the highest waves" is unclear whether it refers to wave heights or wavelengths. Please specify.*

We revised the sentence (lines 614-616):

"*Furthermore, the sensitivity of runup to wave height for Massa's beach slope, combined with the wavelengths associated with the highest waves (70-95 m) is modest, approximately 0.2-0.25 m.*"

As follows:

"*Furthermore, the sensitivity of runup to wave height for Massa's beach slope, calculated using wavelengths ranging between 70 and 95 m (those associated with the highest waves) is modest, approximately 0.2-0.25 m.*"

*6. Lines 654–657: It is unclear why the analysis in this section was conducted only for one location instead of all three sites. The phrase "number of events per year higher than specific values" would benefit from clarification; please indicate what those specific threshold values are.*

With "number of events per year higher than specific values" we intend those values reported in the y axis of the bar coloured subplots.

We modified the proposition (lines 656-657):

"*Furthermore, we determined the number of events per year higher than specific values, clustering the events by decade (Figure 8b and 9b).*"

as follows:

"*Furthermore, we determined the number of events per year (coloured patches) higher than specific values of η and Q (reported in the abscissa), clustering the events by decade (Figure 8b, c, d and 9b, c, d, for Historical, RCP4.5 and RCP8.5 run, respectively).*"

We correct the Caption of Figure 9:

"*a) Cumulative duration of water discharge above the 99.9 %-ile in days per year for HIST (green line), RCP4.5 (red line), RCP8.5 (black line), RCP4.5 and RCP8.5 plus the effect of RSLR (red dashed line and black dashed line, respectively), for the city of Massa. Number of events per year with peak values larger than specific values, grouped by decades for: HIST b), RCP4.5 c) and RCP8.5 d)*"

Removing the part related to RSLR (typing mistake).

In addition, we will include in the Supplementary material the analysis performed for the city of Villanova and Oarsoaldea. We mention it in the main text by adding the following sentence after line 654:

"*both η_TOT(t) and Q(t). (The same analysis for the city of Villanova and Oarsoaldea is reported in the Supplementary Material, Figures from S14 to S17).*"

[Figure]

Fig S14 a) Cumulative duration of total water level above the 99.5 %-ile in days per year for HIST (green line), RCP4.5 (red line), RCP8.5 (black line), RCP4.5 and RCP8.5 plus the effect of RSLR (red dashed line and black dashed line, respectively), for the city of Villanova. Number of events per year with peak values larger than specific values, grouped by decades for: HIST b), RCP4.5 c) and RCP8.5 d)

[Figure]

Fig S15 a) Cumulative duration of water discharge above the 99.9 %-ile in days per year for HIST (green line), RCP4.5 (red line), RCP8.5 (black line), for the city of Villanova. Number of events per year with peak values larger than specific values, grouped by decades for: HIST b), RCP4.5 c) and RCP8.5 d).

[Figure]

Fig S16 a) Cumulative duration of total water level above the 99.5 %-ile in days per year for HIST (green line), RCP4.5 (red line), RCP8.5 (black line), RCP4.5 and RCP8.5 plus the effect of RSLR (red dashed line and black dashed line, respectively), for the city of Oarsoladea. Number of events per year with peak values larger than specific values, grouped by decades for: HIST b), RCP4.5 c) and RCP8.5 d)

[Figure]

Fig S17 a) Cumulative duration of water discharge above the 99.9 %-ile in days per year for HIST (green line), RCP4.5 (red line), RCP8.5 (black line), for the city of Oarsoaldea. Number of events per year with peak values larger than specific values, grouped by decades for: HIST b), RCP4.5 c) and RCP8.5 d).

EGUSPHERE-2025-270
Response to referee 2

We thank the referee for the useful and productive comments. Here, we reply to the raised issues. Referee comments are reported in blue italic font.

**Referee 2**

*This paper provides an example of flood modelling in two coastal cities due to storm surge and river discharge. The paper is well written and is relevant to NHESS. Some uncertainties - e.g. on compound events - are well discussed, but not all. However, before publishing this work, I recommend to:*

*1. provide more elements about the validation of this work*

We would like to thank the referee for this very pertinent comment, which we found particularly useful and motivating. Alongside a similar observation raised by another referee, it has encouraged us to expand the manuscript by including a previously omitted but important part of our work, namely the calibration and validation activities carried out for the different models. These additions help provide a more complete and transparent picture of the modelling chain and its reliability, and we are grateful for the opportunity to improve the manuscript accordingly.

In particular, we plan to add a specific section to the Supplementary Material detailing the calibration/validation procedure for the water level, wave climate and hydrological models. To facilitate readability we uploaded the cal_val.pdf document containing the newly added section to the Supplementary Material.

We intend to modify the main text by expanding the lines from 261 to 263 as follows: "*In this section, we describe the implementation of the three numerical models: WaveWatch III, SHYFEM and LISFLOOD, employed to perform the main part of the downscaling procedure. Each of the models has a particular setup on the basis of the analyzed coastal city. Furthermore, a calibration/validation procedure has been carried out for each of them to have an estimate of their skill to reproduce observed events. The detailed description of the different procedures is reported in the Supplementary Material.*"

For the sake of clarity, we are using the expression "calibration/validation" procedure in a broad sense, and not according to the strict definition involving the use of a dataset to tune the model parameters (calibration), and then another dataset to check the model skill (validation). Due to data availability constraints, we show that the models are able to match observations in case a systematic trial and error calibration procedure was performed (water level model - SHYFEM), and in case the calibrated setup was inherited from a previous study (wave climate model - WWIII), or when it was based on a shared community effort (hydrological model LISFLOOD). The above specification is also included in the cal_val.pdf document.

*2. Assuming a proper validation of the total water levels at a tide gauge and waves offshore, the key uncertainty will result from the use of an empirical formula for the wave setup, which requires a high resolution shallow water bathymetry. It would be important to discuss it as the ambition is to provide a high resolution flood model, as illustrated by the use of a Lidar topographic data.*

We recognize that the use of an empirical formula to calculate the wave runup (Atkinson et al., 2017) is a limitation of the present approach. Specifically, the formula provides the combined value of swash and wave setup derived from offshore wave climate variables ($H_s$ and $L$) and the beach slope. The beach slope was estimated as an average value over the analyzed area by integrating Lidar data with bathymetric information extracted from nautical charts. For the Massa area only, two single beam surveys were available from the years 2012 and 2016. While this introduces a degree of uncertainty due to the alongshore variability of the beach, it also provides an advantage: it allows us to avoid the full dynamical simulation of the wave breaking and swash process, while still capturing the correct order of magnitude for the wave runup estimates. A more advanced approach would be the use of a full coupled hydrodynamic simulation integrating 2D hydrodynamic modelling of the riverine flood wave with the wave and storm surge induced flooding dynamics. At our knowledge, a few models (even with some limitations and expedients, e.g. MIKE, Telemach-Mascaret, FUNWAVE) allow this. We prefer, as a starting point, we have deliberately adopted a simpler approach, also in order to reduce the computational cost. In future developments, we intend to improve our analysis performing coupled hydrodynamic simulations that integrate both wave induced and riverine flood.

We added the following, starting from line 433, to the Section 5.2 Hydrodynamic model:

"*…from the LIDAR dataset, at a resolution of 2 m (Figure 4g, h, i), merging it to information from nautical charts, except for the city of Massa, where two single beam surveys were available for the years 2012 and 2017.*"

We also expand the Discussion section including the following starting from line 678:

"*The use of an empirical formula to calculate the wave runup (Atkinson et al., 2017), while avoiding us to fully simulate the dynamical swash process and getting at least the order of magnitude of runup values, introduces uncertainties due to the degree of alongshore variability of the beach or due to the reduced knowledge of the underwater bathymetry. Indeed, for the city of Massa two bathymetric surveys were available (2012 and 2016), but for Villanova the submerged part of the domain principally comes from nautical charts. Specific efforts to recover the detailed bathymetry of the area are recommended to make the resolution of the hydrodynamic domain as uniform as possible.*"

*3. The context about sea-level rise is not well explained; it would be important to discuss what we know from sea-level rise, e.g. commitments over decades to centuries, the fact that meters of sea level rise can not be avoided on the long term and that this could happen earlier than projected in case of an ice sheet collapse - see e.g. the IPCC report WG1 published in 2021. Based on this review, I recommend to reconsider the statements starting lines 605 regarding future flooding in Massa.*

We agree the statement, as it was phrased, could convey a distorted idea of the results obtained. We intend to modify the sentence at lines 605-606:

"*Future coastal floods in Massa do not show significant variations in terms of event magnitude compared to the Historical period.*"

as follows:

"*If we only look at the return period values of total water level for the city of Massa, we do not observe significant variations in terms of event magnitude compared to the Historical period.*"

Lines 618-619:

"*Actually, the main driver in producing significant differences in flooded volume is the RSLR (Table 3), which causes the storm surge to penetrate farther inland, resulting in larger flooded volumes (Table 7).*"

were substituted with the following:

"*Actually, the main driver behind the significant differences in flooded volume is the Relative Sea Level Rise (RSLR) (Table 3), which allows storm surges to penetrate farther inland, resulting in larger flooded volumes (Table 7). This finding is consistent with the conclusions of the IPCC Sixth Assessment Report (IPCC, 2023b), which states that regional sea level change will be the primary factor contributing to a substantial increase in the frequency of extreme still water levels over the next century, even assuming other contributors to extreme sea levels to remain constant. Therefore, all uncertainties in sea level rise projections can significantly affect the flood extension and volume associated with extreme events. In addition, the projected sea level rise by the end of the century could be significantly higher if the less likely, but still plausible, ice-sheet-related dynamics were to occur (IPCC, 2023b; IPCC, 2014).*"

We will also integrate the bibliography:

Intergovernmental Panel on Climate Change (IPCC). (2023a). Climate Change 2022 – Impacts, Adaptation and Vulnerability: Working Group II Contribution to the Sixth Assessment Report of the Intergovernmental Panel on Climate Change. Cambridge University Press. https://doi.org/10.1017/9781009325844

Intergovernmental Panel on Climate Change (IPCC). (2023b). Climate Change 2022 – The Physical Science Basis: Working Group I Contribution to the Sixth Assessment Report of the Intergovernmental Panel on Climate Change. Cambridge University Press. https://doi.org/10.1017/9781009157896

And we plan to add the references in lines 49:

"*worldwide, are among the most vulnerable areas to these events (IPCC, 2023a; Vitousek et al., 2017;*"

and 61:

"*...year coastal flood could increase by about 20% in the medium to long term (IPCC, 2023a). By 2100, the..*"

*4. The data used on sea-level projections are not clearly presented (no unit in Table 3, no source associated to the table). This should be precisely clarified.*

We report here the updated version of Table 3.

| $\Delta\eta_{\mathrm{RSLR}}$ [m] | Massa | Villanova | Oarsoaldea |
|---|---|---|---|
| RCP4.5 2011-2060 | 0.150 (-0.036, +0.05) | 0.15 (-0.044, +0.056) | 0.192 (-0.057, +0.061) |
| RCP4.5 2051-2100 | 0.351 (-0.096, 0.131) | 0.349 (-0.105, +0.138) | 0.412 (-0.155, +0.150) |
| RCP8.5 2011-2060 | 0.168 (-0.042, +0.057) | 0.173 (-0.053, +0.062) | 0.229 (-0.079, +0.073) |
| RCP8.5 2051-2100 | 0.464 (-0.137, +0.173) | 0.458 (-0.136, +0.185) | 0.537 (-0.208, +0.203) |

Table 3. Values of RSLR referred to the RCP4.5 and RCP8.5 scenarios, averaged over the reference period, for the analyzed cities. All values are given with respect to the 1985-2005 reference period. Data extracted from Vousdoukas et al. 2016a ( RCP4.5 data) and Vousdoukas et al. 2016b (RCP8.5 data).

*5. Overall the paper would have more impact with a one or a few clear key messages that could be taken forward - e.g. that precise flood hazard assessment is required to assess future sea level rise impacts in coastal cities, that this can not be achieved at broad scale yet (no global Lidar DEM) and requires local assessments;*

We intend to modify the concluding part of the Abstract

"*This work demonstrates the capability of the integrated framework to address climate change impacts at urban scales, providing valuable insights for the development of localized adaptation strategies.*"

as follows:

"*This work demonstrates the capability of an integrated modeling framework to address climate change impacts at the urban scale. Local-scale modeling is essential: accurate flood hazard assessment in coastal cities requires high-resolution simulations to capture the influence of local topography and infrastructure, especially where global DEMs are inadequate. By linking climate projections to urban flood impacts, the framework enables a consistent evaluation of future extremes, sea level rise, and their interaction. A further key message of this study is the need to generate actionable insights to support the development of targeted and site-specific adaptation strategies. Adaptation must be tailored: only by quantifying future extremes and exposure is it possible to design effective, place-based responses.*"

To reinforce the overall message, we plan to reiterate these key points also in the Conclusions section, highlighting their relevance for future applications and decision-making.

Addendum to the Supplementary Material

**Calibration/validation procedure for the employed models**

For the sake of clarity, we are using the expression "calibration/validation" procedure in a broad sense, and not according to the strict definition involving the use of a dataset to tune the model parameters (calibration), and then another dataset to check the model skill (validation). Due to data constraints, we show that the models are able to match observations in case a systematic trial and error calibration procedure was performed (water level model - SHYFEM), and in case the calibrated setup was inherited from a previous study (wave climate model - WWIII), or when it was based on a shared community effort (hydrological model LISFLOOD). The above specification is also included in the cal_val.pdf document.

**1 Calibration and validation procedure for the water level and wave climate models**

**1.1 Water level model (SHYFEM)**

The water level model was calibrated and validated against observed data close to the city of Massa and Villanova, in the Mediterranean Sea, and Oarsoaldea, in the Atlantic Ocean. A trial and error procedure was used to identify the combination of wind drag and bottom friction coefficients leading to the best representation of the residual $\eta$ (that is when the tidal components are removed). Modelled $\eta$ values were compared to tidal gauge observations for the year 2012 for the city of Massa, and for the year 2020 for the city of Oarsoaldea. Wind and atmospheric pressure data at 3-hourly frequency from the Integrated Forecasting System (IFS) of the European Centre for Medium Range Weather Forecast (ECMWF) were used as atmospheric forcing for the calibration runs.

Model skill was estimated computing the correlation coefficient (CORR), the root mean squared error (RMSE) and BIAS for the whole time series and for $\eta$ values higher than 0.25 m for Oarsoaldea and higher than 0.15 m for Massa roughly indicating moderate storm surge events (Marcos et al., 2011).

Several calibration runs were performed for the city of Oarsoaldea (tidal gauge located at Lon = -1.931 E; Lat = 43.321 N) with the best result given by: CORR = 0.81, RMSE = 0.08 m, BIAS = -0.06 m for the whole time series; RMSE = 0.09 m; BIAS = -0.01 for $\eta$ > 0.25 m. Figure S8a and b presents the observed and modelled residual $\eta$ from 9 calibration runs (m1 to m9), focusing on periods with residual water levels peaks reaching up to 47 cm. Despite the moderate variability in the computed values, most time series satisfactorily reproduce the observed signal, particularly during peak events. Results from simulations m8 correspond to the best calibration run.

[Figure]

**Fig S8** a) Time evolution of the observed and modelled residual water level for 9 calibration runs for December 2020 for the Oarsoaldea tidal gauge. b) Time evolution of the observed and modelled residual water level for 9 calibration runs from mid-September to mid November 2020 for the Oarsoaldea tidal gauge.

Same approach was employed for the city of Massa (tidal gauge located at Lon = 9.8576 E; Lat = 44.0966 N) for which the best calibration run gave the following results: CORR = 0.79, RMSE = 0.05 m, BIAS = 0.01 m for the whole time series; RMSE = 0.06 m, BIAS = 0.04 for $\eta > 0.15$ m. Figure S9a and b show the observed and modelled residuals for 9 calibration runs (m1 to m9). mX corresponds to the best calibration run.

[Figure]

**Fig S9** a) Time evolution of the observed and modelled residual water level for 9 calibration runs for October and a few days of November 2012 for the Massa tidal gauge. b) Time evolution of the observed and modelled residual water level for 9 calibration runs for the last part of November and December 2012 for the Massa tidal gauge.

Even for the other Mediterranean city, Villanova, the same procedure has been followed, obtaining similar estimation of the model prediction accuracy.

**1.2 Wave climate model (WWIII)**

The wave model employed was already calibrated and validated by Vannucchi et al. (2021), but for a numerical mesh including only the Mediterranean Sea and different forcing fields based on a dynamic downscaling of the ERA-5 dataset (Hersbach et al., 2020). In the present work, the same calibrated parameters were adopted under the assumption that they remain valid outside the Mediterranean region as well.

For the present study, we report the model skill in reproducing the $H_s$ with respect to observations from the Gorgona buoy (Lon = 9.96 E; Lat = 43.57 N), located in the Mediterranean Sea in the Tuscany Archipelago, and for one located close to Oarsoaldea, in the Atlantic Ocean (Lon = 1.89 E; Lat = 43.37 N). Even if the wave climate is not employed to simulate floods in Oarsoaldea, we report it for completeness of information.

For the Gorgona buoy, considering a period from 2009 to 2018, we obtained the following values: CORR = 0.64, RMSE = 0.65 m, BIAS = -0.11 m. Furthermore, to better quantify the model skill with respect to the observed quantity, the standard deviation (STD) is equal to 0.77 m and maximum $H_s$ ranging between 5 m and almost 8 m. In addition, we report the statistics obtained by Vannucchi et al. (2021) comparing the modelled values against the same buoy data: CORR = 0.92 RMSE = 0.29 m BIAS = -0.09 m. In Figure S10a and b, the time evolution of the observed and modelled $H_s$ for the Gorgona buoy, for two periods characterized by intense wave climate, is reported.

[Figure]

**Fig S10** Time evolution of the observed and modelled significant wave height for the Gorgona buoy for three periods: a) from December 2009 to April 2010; b) from November 2011 to February 2012.

For the Oarsoaldea buoy, we have data from 2010 to 2012, and the model skill is summarized by the following values: CORR = 0.84, RMSE = 0.51 m, BIAS = -0.10 m, STD = 0.85 m. Maximum values range between 5 m and 8 m. Figures S11a and b, report the result for the Oarsoaldea buoy.

[Figure]

**Fig S11** Time evolution of the observed and modelled significant wave height for the Oarsoaldea buoy for three periods: a) from November 2009 to February 2010; b) from November 2014 to February 2015.

Furthermore, Figure S12a and b report the QQ-plots for the observed (x-axis) and modelled (y-axis) $H_s$ for the Gorgona and Oarsoaldea buoy, respectively. In both cases it is shown that the model tends to underestimate the largest wave heights.

[Figure]

**Fig S12** QQ-plots of the observed and modelled significant wave height for the Gorgona buoy a) and Oarsoaldea buoy b).

**2 Calibration and validation procedure for the hydrological model (LISFLOOD)**

LISFLOOD hydrological model was implemented for the three study areas (Massa, Villanova and Oarsoaldea) with a set of standard parameters selected within the values suggested from the routine calibration activities executed in the framework of the European Flood Awareness System (EFAS) for European basins, or worldwide for the Global Flood Awareness System (GloFAS). Both systems use LISFLOOD in the operational modelling chain for flood hazard forecast (Ziese et al., 2019, Hirpa et al 2018, Dottori et al., 2022).

For the present application the main calibration parameters, i.e. the ones that control the infiltration capacity and runoff generation, were set to b_Xinjang = 0.7 and power_pre flow = 3.5. In order to evaluate the performance of the model with this configuration, a set of focused simulations were performed on the Frigido river basin, where a water level gauge is present at the station of Canevara (located in the upper part of the basin, with a total catchment area of approx. 50 km$^2$). The simulation period goes from 2018 to 2021, when a valid stage-discharge relationship is available. Hourly precipitation time series required as meteorological forcing were retrieved from two ground-based gauges located in the upstream basin. Both water levels and precipitation data were provided by Tuscany Region Hydrological Service, www.sir.toscana.it). The other meteorological variables required for the LISVAP preprocessing step (incoming solar radiation, air temperature, dewpoint temperature, wind speed) were extracted from ERA5-Land product (Muñoz Sabater, J., 2019). An outlet point was added in LISFLOOD settings corresponding to the hydrometric gauge site and the hourly discharge time series obtained from the simulations were compared with the observed values.

Figure S13 shows the modelled and observed hydrographs for several events in the simulated period. The results show that the LISFLOOD model is in most cases capable of representing the flood peaks with adequate accuracy, whereas a general underestimation of baseflow and a steeper descendant limb of the hydrograph result from the model compared to the observed values. A more accurate baseflow estimation

can be obtained with an improved groundwater representation and further calibration of the slow-response parameters. For the scope of the present study, that is focused on extreme events, the fair simulation of peak discharges was considered satisfactory, also considering the uncertainties related to spatial precipitation patterns and the relatively scarce data available in small basins.

These simulations, performed using in-situ rainfall observations, confirm the capabilities of the LISFLOOD model to adequately reproduce the hydrological processes in the study basin. However, it is necessary to point out that when the model is applied with different types of forcing, as in the case of global or regional climate models, additional uncertainties, biases and sources of errors may arise and clearly affect the accuracy of the hydrologic simulation. For example, we expect an underestimation of peak rainfall intensities when using gridded data with coarse spatial and temporal resolution.

Since the present work is dedicated to the general presentation of the modelling chain and the relative comparison between different climatologic scenarios, we did not address these aspects in detail but they are the object of further ongoing activities.

[Figure]

**Fig S13** Modelled (blue line) and observed (black dots) hourly discharge for several events in the period 2018-2021 at Canevara hydrometric station on the Frigido river.

**References**

Dottori, F., Alfieri, L., Bianchi, A., Skoien, J., and Salamon, P.: A new dataset of river flood hazard maps for Europe and the Mediterranean Basin, Earth Syst. Sci. Data, 14, 1549–1569, https://doi.org/10.5194/essd-14-1549-2022, 2022.

Hersbach, H., Bell, B., Berrisford, P., Hirahara, S., Horányi, A., Muñoz-Sabater, J., ... & Thépaut, J. N. (2020). The ERA5 global reanalysis. Quarterly journal of the royal meteorological society, 146(730), 1999-2049.

Muñoz Sabater, J. (2019): ERA5-Land hourly data from 1950 to present. Copernicus Climate Change Service (C3S) Climate Data Store (CDS). DOI: 10.24381/cds.e2161bac (Accessed on DD-MMM-YYYY)

Vannucchi, V.; Taddei, S.; Capecchi, V.; Bendoni, M.; Brandini, C.; Dynamical Downscaling of ERA5 Data on the North-Western Mediterranean Sea: From Atmosphere to High-Resolution Coastal Wave Climate. J. Mar. Sci. Eng. 2021, 9, 208. https://doi.org/10.3390/jmse9020208

Ziese, M., Garcia, R., Dottori, F., Salamon, P., Schweim, C. et al., *EFAS upgrade for the extended model domain – Technical documentation*, Publications Office, 2019, https://doi.org/10.2760/806324